# Mechanism for differential recruitment of orbitostriatal transmission during actions and outcomes following chronic alcohol exposure

Rafael Renteria[1], Christian Cazares[2], Emily T Baltz[2], Drew C Schreiner[1], Ege A Yalcinbas[2], Thomas Steinkellner[3], Thomas S Hnasko[2,3,4], Christina M Gremel[1,2]*

[1]Department of Psychology, University of California San Diego, San Diego, United States; [2]The Neurosciences Graduate Program, University of California San Diego, San Diego, United States; [3]Department of Neurosciences, University of California San Diego, San Diego, United States; [4]Research Service, VA San Diego Healthcare System, San Diego, United States

**Abstract** Psychiatric disease often produces symptoms that have divergent effects on neural activity. For example, in drug dependence, dysfunctional value-based decision-making and compulsive-like actions have been linked to hypo- and hyperactivity of orbital frontal cortex (OFC)-basal ganglia circuits, respectively; however, the underlying mechanisms are unknown. Here we show that alcohol-exposed mice have enhanced activity in OFC terminals in dorsal striatum (OFC-DS) associated with actions, but reduced activity of the same terminals during periods of outcome retrieval, corresponding with a loss of outcome control over decision-making. Disrupted OFC-DS terminal activity was due to a dysfunction of dopamine-type 1 receptors on spiny projection neurons (D1R SPNs) that resulted in increased retrograde endocannabinoid signaling at OFC-D1R SPN synapses reducing OFC-DS transmission. Blocking CB1 receptors restored OFC-DS activity in vivo and rescued outcome-based control over decision-making. These findings demonstrate a circuit-, synapse-, and computation-specific mechanism gating OFC activity in alcohol-exposed mice.

*For correspondence: cgremel@ucsd.edu

Competing interests: The authors declare that no competing interests exist.

## Introduction

The DSM-5 categorizes alcohol as one of the few drugs whose dependence can produce neurocognitive disorders such as alcohol-related dementia (*American Psychiatric Association, 2013*). Thus, treatment of alcohol dependence would benefit from understanding mechanisms producing impaired cognition. Value-based decision-making is one such cognitive process, with its disruption contributing to poor daily function, that when combined with the emergence of compulsive control contributes to excessive drug-seeking and relapse (*Belin et al., 2013*; *Everitt and Robbins, 2016*; *Gremel and Lovinger, 2017*; *Hogarth et al., 2013*; *Lüscher et al., 2020*). Recently, there has been renewed focus on orbital frontal cortex (OFC) circuits in substance use disorder and related behaviors (*Everitt and Robbins, 2016*; *Lüscher et al., 2020*; *Moorman, 2018*; *Schoenbaum et al., 2016*; *Schoenbaum et al., 2006*), as they play a key role in computations related to decision-making and the evaluation of outcomes (*Fellows, 2007*; *Stalnaker et al., 2015*; *Wallis, 2007*). However, a mechanistic understanding of how drug dependence disrupts the ability of OFC and its downstream circuits to represent information necessary for appropriate control over decision-making is lacking.

Indeed, drug dependence in general has been widely associated with reduced OFC activity and output (*Catafau et al., 1999*; *Lüscher et al., 2020*; *Schoenbaum et al., 2016*; *Schoenbaum et al., 2006*; *Volkow et al., 1997*; *Volkow and Fowler, 1994*). In alcohol-dependent individuals, reduced OFC activity correlates with aberrant value-based decision-making (*Boettiger et al., 2007*; *Duka et al., 2011*; *Forbes et al., 2014*). However, OFC hyperactivity has been observed in alcohol dependence in response to predictive information (*Hermann et al., 2006*; *Reinhard et al., 2015*; *Tapert et al., 2003*) and during approach behaviors (*Ernst et al., 2014*). In concert, both rodent and non-human primate models of alcohol dependence have identified long-lasting changes to OFC neuron excitability, structure, and transmission (*Nimitvilai et al., 2017*; *Nimitvilai et al., 2016*; *Renteria et al., 2018*). Thus, it seems likely that alcohol dependence differentially alters the contribution of OFC circuits and their downstream controllers to varied decision-making processes, thereby enhancing some computations while reducing others, leading to alterations in behavioral control.

One such circuit is OFC excitatory projections into basal ganglia, with synapses onto direct and indirect pathways in the dorsal striatum (DS) (*Renteria et al., 2018*; *Wall et al., 2013*). Excitatory transmission from OFC-DS projections carries information that may support value-based decision-making, as inhibition of transmission results in habitual control (*Gremel et al., 2016*). We recently found that chronic alcohol exposure selectively attenuated output of this circuit, with a long-lasting reduction in OFC-DS transmission onto dopamine-type 1 (D1) spiny projection neurons (SPNs) of the direct pathway (*Renteria et al., 2018*). In addition, chronic alcohol exposure produces a loss of value-based decision-making (*Corbit et al., 2012*; *Dickinson et al., 2002*; *Renteria et al., 2018*) and reductions in associated OFC activity (*Catafau et al., 1999*; *Lüscher et al., 2020*; *Schoenbaum et al., 2016*; *Schoenbaum et al., 2006*; *Volkow et al., 1997*; *Volkow and Fowler, 1994*). Indeed, this behavioral phenotype was rescued by increasing OFC activity (*Renteria et al., 2018*). Together, these findings suggest that alcohol dependence may result in a reduced communication of value-related information from OFC into basal ganglia circuits to support decision-making.

However, dependence has also been associated with compulsive phenotypes (*Everitt and Robbins, 2016*; *Everitt and Robbins, 2005*; *Lüscher et al., 2020*), and hyperactivity of OFC-DS circuits has been implicated in obsessive compulsive disorder (OCD) (*Lüscher et al., 2020*; *Milad and Rauch, 2012*; *Pauls et al., 2014*; *Robbins et al., 2019*), as well as animal models of compulsive (*Ahmari et al., 2013*) and compulsive behaviors (*Pascoli et al., 2018*; *Pascoli et al., 2015*). Indeed, increased OFC terminal activity was observed in mice when they self-stimulated ventral tegmental area dopamine neurons, despite the presence of a punishing foot-shock (*Pascoli et al., 2018*). This corresponded to postsynaptic potentiation of OFC-DS transmission (*Pascoli et al., 2018*), suggesting that increased transmission from OFC into basal ganglia circuits contributes to the emergence of compulsive actions. Thus, OFC-DS circuits are poised to regulate both compulsive action and value information, computations that are both implicated in substance use disorder and may be differentially altered in psychiatric disease.

Here we examine mechanisms underlying how drug dependence affects OFC-DS circuit information representation and its control over decision-making. Using a well-established model of alcohol dependence in mice, we find that chronic alcohol exposure selectively enhances OFC-DS activity associated with action while reducing OFC-DS activity during periods associated with outcome retrieval through downregulation of D1 receptor function in SPNs and enhanced endocannabinoid (eCB) signaling at OFC-D1 SPN synapses in the DS. Restoring OFC-DS activity rescues value-based decision-making. Our data has important implications for hypotheses regarding compulsive and habitual phenotypes observed in substance use disorder.

## Results

### Altered encoding of OFC-DS terminal activity following chronic alcohol exposure

To examine whether alcohol dependence alters information sent to dorsal striatum, we used fiber photometry to measure calcium activity of OFC terminals in DS during self-initiated value-based decision-making. We utilized a well-validated and commonly used mouse model of alcohol dependence, chronic intermittent ethanol (CIE) vapor exposure (*Becker, 1994*; *Becker and Lopez, 2004*;

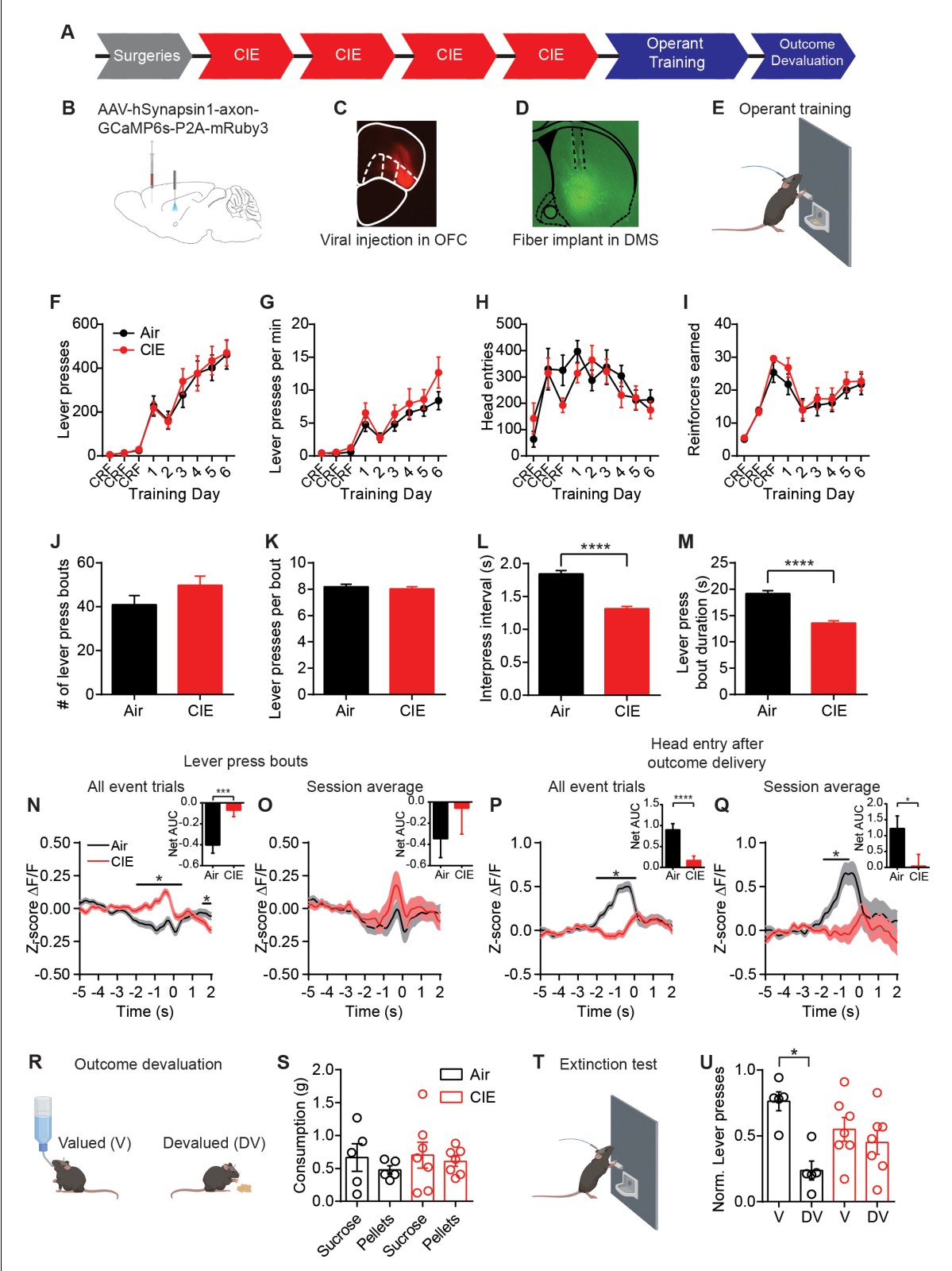

**Figure 1.** CIE-induced alterations in OFC terminal activity in the DS during instrumental responding. (**A**) Experimental timeline that includes surgeries followed by four cycles of CIE exposure, operant training, and outcome devaluation. (**B**) Schematic of viral injections in the OFC and fiber implant in the DS. (**C**) Example viral expression in OFC. (**D**) Example OFC terminal expression and fiber placement in the DS. (**E**) Lever press training for a food pellet under random ratio schedule of reinforcement. (**F**) Lever presses, (**G**) response rate, (**H**) head entries, and (**I**) reinforcers earned during training in Air

*Figure 1 continued on next page*

*Figure 1 continued*

(n = 10 mice, three vapor cohorts) and CIE (n = 12 mice, three vapor cohorts) mice. (J) Average number of lever press bouts in Air and CIE mice. (K) Average number of lever presses per bout. (L) Average inter-press interval within lever press bouts. (M) Average bout duration. (N) Z-score of ΔF/F GCaMP6s traces recorded in OFC terminals during lever press bouts including all event trials and net area under the curve (AUC) of Z-score GCaMP6s signal during lever press bouts. (O) Z-score of ΔF/F GCaMP6s traces during lever press bouts including session averages and net area under the curve (AUC) of Z-score GCaMP6s signal during lever press bouts. (P) Z-score of ΔF/F GCaMP6s trace during the first head entry after outcome delivery including all event trials and net AUC of Z-score GCaMP6s signal during first head entry after outcome delivery. (Q) Z-score of ΔF/F GCaMP6s trace during the first head entry after outcome delivery including session averages and net AUC of Z-score GCaMP6s signal during first head entry after outcome delivery. (R) Schematic of outcome devaluation procedure. (S) Consumption of sucrose and pellets in Air (n = 5 mice) and CIE (n = 7 mice) mice during 1 hr free feed period. (T) Schematic of extinction test following free feed period. (U) Distribution of lever presses across valued (V) and devalued (DV) days. Data points and bar graphs represent mean ± SEM. Bonferroni-corrected post hoc test of repeated-measures ANOVA, two-sided FDR-corrected permutation test, or Kolmogorov–Smirnov test, *p≤0.05, ****p≤0.0001.

The online version of this article includes the following source data and figure supplement(s) for figure 1:

Source data 1. CIE-induced alterations in OFC terminal activity in the DS during instrumental responding source data.
Figure supplement 1. CIE-induced alterations in OFC terminal activity in the DS during instrumental responding, devaluation testing, and histological placements.
Figure supplement 1—source data 1. *Figure 1—figure supplement 1* CIE-induced alterations in OFC terminal activity in the DS during instrumental responding, devaluation testing, and histological placements source data 1.
Figure supplement 1—source data 2. *Figure 1—figure supplement 1* CIE-induced alterations in OFC terminal activity in the DS during instrumental responding, devaluation testing, and histological placements source data 2.

*Griffin et al., 2009*; *Lopez and Becker, 2005*; *Figure 1A*). Performed in three cohort replications, mice (Air n = 10, CIE n = 12) were exposed to ethanol vapor for 16 hr each day for four consecutive days followed by a 3 day withdrawal. This cycle of ethanol vapor and withdrawal was repeated for four consecutive weeks and resulted in blood ethanol concentrations of 33.70 ± 2.24 mM (mean ± SEM) (see Materials and methods). Prior to CIE procedures, mice were injected with AAV-hSynap-sin1-axon-GCaMP6s-P2A-mRuby3 in the OFC and implanted with a fiber targeted to the medial DS to record OFC terminal activity (see Materials and methods) (*Figure 1B–D*). Alcohol withdrawal has been delineated into two differing stages: an acute withdrawal period lasting 2–3 days and longer protracted withdrawal period that can last three or more months (*American Psychiatric Association, 2013*; *Heilig et al., 2010*). To avoid effects of acute withdrawal and examine potential alterations in the protracted withdrawal period, mice were food restricted and 5 days after the last vapor exposure were trained to lever press for a food reward under a random ratio schedule of reinforcement to bias value-based decision-making, measured as a sensitivity to outcome devaluation (*Dickinson et al., 1983*; *Gremel and Costa, 2013*; *Figure 1E*).

Chronic alcohol exposure did not prevent mice from learning to lever press for food. Air control mice and CIE-exposed mice made the same number of lever presses (repeated-measures ANOVA: main effect of Training day: $F_{(8, 160)}=43.10$, p<0.0001; no main effect of CIE exposure or interaction: Fs < 0.16) (*Figure 1F*) and pressed the lever at roughly equal rates (repeated-measures ANOVA: main effect of Training day: $F_{(8, 160)}=27.42$, p<0.0001; no main effect of treatment or interaction: Fs < 1.22) (*Figure 1G*). The two groups made similar number of head entries across training (repeated-measures ANOVA: main effect of Training day: $F_{(8, 160)}=5.92$, p<0.0001; no main effect of CIE exposure or interaction: Fs < 1.29) (*Figure 1H*) and earned similar numbers of rewards (repeated-measures ANOVA: main effect of Training day: $F_{(8, 160)}=19.42$, p<0.0001; no main effect of treatment or interaction: Fs < 0.60) (*Figure 1I*). As mice self-initiate, self-terminate their own behavior, it appeared they often made multiple lever presses in close succession. We used bout analyses to examine whether mice organized their lever presses into distinguishable patterns (See Materials and methods). Mice grouped their lever presses into distinct bouts, and Air and CIE mice had similar numbers of bouts (Air: 40.92 ± 4.15, n = 37; CIE: 49.78 ± 4.17, n = 37; non-parametric two-sample Kolmogorov–Smirnov [KS] test: D = 0.30, p=0.07) (*Figure 1J*) with similar numbers of lever presses per bout (Air: 8.19 ± 0.18, n = 1514; CIE: 8.032 ± 0.16, n = 1842; KS test: D = 0.04, p=0.14) (*Figure 1K*). However, Air and CIE mice differentially executed these bouts, with CIE mice having shorter intervals between lever presses within a bout (inter-press interval, Air: 1.84 s ± 0.05, n = 1514; CIE: 1.31 s ± 0.04, n = 1842; KS test: D = 0.17, p<0.0001) (*Figure 1L*), resulting in shorter bout durations (Air: 19.15 s ± 0.59, n = 1514; CIE: 13.59 s ± 0.41, n = 1842; KS test: D = 0.14,

p<0.0001) (*Figure 1M*). Thus, chronic alcohol exposure led to a change in action execution, manifested as faster completion of lever press bouts.

When we examined changes in z-scored (z-score ΔF/F0) OFC-DS terminal $Ca^{2+}$ activity during decision-making, we observed modulation of calcium transients during action epochs (−2 s to 2 s after the onset of a lever press bout compared to −5 s to −2 s baseline period) and outcome retrieval epochs (−2 s to 2 s around head entry into food receptacle following reward delivery compared to −5 s to −2 s baseline period). As mice can readily see the food reward prior to making a head entry, this outcome retrieval period encompasses reward perception, retrieval, and potentially consumption. We examined whether modulations of calcium transients were differentially affected by chronic alcohol exposure. Photometry data was analyzed two ways. First, z-scored ΔF/F0 traces were combined across all mice within a group (all event trials). This was done to preserve the variance seen within a subject. Second, we averaged these z-scored ΔF/F0 traces for a given animal session and then averaged these traces across mice within a group (session average). This examines between-mouse variability but does not preserve within-subject variability. When we examined all lever presses, we saw CIE led to a slight yet significant increase in OFC-DS $Ca^{2+}$ activity prior to the onset of a lever press, both when we examined lever presses collapsed across animals as well as when we examined session averages per mouse (see Materials and methods) (*Figure 1—figure supplement 1C,G*). Given mice organized their lever presses into bouts in a self-paced nature and can make multiple lever presses within the analysis window prior to lever press onset, we next examined whether this CIE-induced increase in OFC-DS modulation was evident in the first lever press of a bout. When all event trials where collapsed across mice, we found increased OFC-DS activity at the onset of lever press bout (false discovery rate [FDR]-corrected two-sided permutation test, all event trials: Air: n = 1363, CIE n = 1650, p<0.05), and net area under the curve (AUC) analyses showed a significant difference between Air and CIE mice (Air: −0.40 ± 0.08, n = 1363; CIE: −0.05 ± 0.06, n = 1650; unpaired t-test: $t_{3011}$ = 3.70, p<0.001) (*Figure 1N*). However, analyses of average session per mouse activity at the onset of a lever press bout did not show a significant increase (session average: Air: n = 35; CIE: n = 36, p>0.05) or difference in AUC (Air: −0.35 ± 0.18, n = 35; CIE: −0.06 ± 0.24, n = 36; unpaired t-test: $t_{69}$ = 0.94, p=0.35) (*Figure 1O*). Together this suggests that CIE may lead to slight increases in OFC terminal activity during lever-press-related behavior.

In contrast, when we examined OFC-DS activity during the period associated with outcome retrieval following reward delivery (first head entry after outcome delivery), we found the opposite pattern. There, in both all event and session average analyses, Air controls showed a large significant increase in OFC-DS terminal activity, whereas in CIE mice, this was significantly blunted (FDR-corrected two-sided permutation test, all event trials: Air: n = 521, CIE: n = 682; session average: Air: n = 34; CIE: n = 38) (*Figure 1P,Q*). Furthermore, there was an overall decrease in the net AUC in CIE mice compared to Air mice (all event trials: Air: 1.00 ± 0.16, n = 521; CIE: 0.24 ± 0.11, n = 682; unpaired t-test: $t_{1201}$ = 4.12, p<0.0001; session average: Air: 1.22 ± 0.4, n = 34; CIE: 0.04 ± 0.37, n = 38; unpaired t-test: $t_{70}$ = 2.18, p<0.05) (*Figure 1P,Q*). Since the sound associated with pellet delivery could potentially contribute to activity modulation, we examined OFC-DS terminal activity in relation to outcome delivery (independent of any checking behavior). Again, we found greater modulation within 1 s of outcome delivery in Air mice than in CIE mice (*Figure 1—figure supplement 1D,H*). Furthermore, the increase in OFC-DS activity in Air, and to a lesser extent CIE mice, was selective to head entries following outcome delivery when compared to non-rewarded head entries (*Figure 1—figure supplement 1E,F*). Together, these findings suggest that CIE exposure may lead to slightly enhanced OFC-DS terminal activity associated with actions, but substantially less OFC-DS terminal activity during the period associated with outcome perception and/or retrieval.

Given the necessity of OFC-DS pathway function for successful outcome devaluation (*Gremel et al., 2016*), the substantial reduction in OFC-DS activity observed in CIE during periods associated with outcome retrieval suggests value-based decision-making may be impaired in these CIE mice. Thus, we used sensory-specific satiety to induce outcome devaluation (*Figure 1R–U*, see Materials and methods), where reductions in lever pressing on the devalued compared to valued day are indicative of outcome devaluation. Pre-planned comparisons on normalized data to account for differences in response rates between individual mice showed that Air reduced responding following outcome devaluation, while CIE mice did less so (repeated-measures ANOVA: main effect of valuation state: $F_{(1, 10)}$=6.36, p<0.05; no main effect of CIE exposure or interaction: Fs < 2.98; pre-planned corrected comparisons between Valued and Devalued days: Air p<0.02; CIE p>0.6)

(*Figure 1U*, *Figure 1—figure supplement 1*). Indeed, Air mice differentially distributed lever pressing between devalued and valued days, and CIE mice did not (one-sample paired t-test vs. 0.5; Air-$t_4$ = 3.68, p<0.05; CIE $t_6$ = 0.54, p=0.30). As previous data found enhancing OFC activity was sufficient to rescue the deficits in outcome devaluation in CIE mice (*Renteria et al., 2018*), the current data suggest that their inability to update outcome value following devaluation may be a result of the dampened OFC-DS activity during outcome retrieval epochs (*Figure 1P,Q*).

## Chronic alcohol exposure does not alter CB1 receptor function or eCB long-term plasticity

The reduced calcium activity in OFC-DS terminals is suggestive of a decrease in presynaptic transmission. In dorsal striatum, OFC terminal synapse onto both direct and indirect output pathways of the basal ganglia (*Wall et al., 2013*). However, we previously found CIE results in a selective and long-lasting decrease in presynaptic OFC transmission onto D1 SPNs of the direct pathway, but not dopamine type-2 (D2) SPNs of the indirect pathway (*Renteria et al., 2018*). This reduction in OFC-D1 SPN transmission was stable and persistent across 3 weeks of chronic withdrawal (*Renteria et al., 2018*). As an OFC-DS projection neuron most likely sends collaterals to both D1 SPNs and D2 SPNs (*Wall et al., 2013*), the cell-type-specific decrease in presynaptic transmission may be mediated by a CIE-induced alteration in eCB signaling (*Henderson-Redmond et al., 2016*; *Parsons and Hurd, 2015*; *Pava and Woodward, 2012*). eCBs are produced and released from the postsynaptic cell and act as a retrograde signal to activate presynaptic cannabinoid type 1 (CB1) receptors, resulting in an inhibition of presynaptic calcium and a subsequent decrease in neurotransmitter release (*Castillo et al., 2012*; *Gerdeman et al., 2002*; *Heifets and Castillo, 2009*; *Kano et al., 2009*; *Kreitzer and Regehr, 2001*; *Lovinger, 2008*). In the dorsal striatum, CB1 receptors have been implicated in habitual action control (*Hilário et al., 2007*; *Nazzaro et al., 2012*), and activation of CB1 receptors on OFC terminals in DS has been shown to disrupt sensitivity to changes in expected outcome value (*Gremel et al., 2016*).

To identify CIE-induced mechanisms by which OFC transmission is selectively decreased to D1 SPNs, nine cohort replicates of mice were exposed to four rounds of CIE as previously described (*Figure 1A*), and electrophysiological recordings were performed within 3–21 days of withdrawal (*Figure 2A*). We have previously shown that the decrease in OFC transmission to D1 SPNs persists in protracted withdrawal and is present for up to 21 days post-CIE procedures (see Supplementary Figure 2 in *Renteria et al., 2018*). Prior to CIE, D1-tdTomato or D1DR-Cre mice were injected with AAV5-CamKIIa-GFP-Cre and a Cre-dependent channelrhodopsin (ChR2) (AAV5-Ef1a-DIO-ChR2-eYFP; UNC viral vector core) in OFC to limit ChR2 expression to CamKIIa expressing excitatory projection neurons (*Gremel et al., 2016*; *Gremel and Costa, 2013*; *Renteria et al., 2018*; *Tye et al., 2011*; *Figure 2B,C*). D1DR-Cre mice received an additional injection of AAV5-Flex-tdTomato in the DS to identify D1-expressing SPNs (*Figure 2D*). Whole-cell patch clamp recordings were performed on tdTomato expressing D1 SPNs in the DS and optically evoked EPSCs were recorded by activating ChR2 expressing OFC terminals with 470 nm light in the DS (*Figure 2B*).

We first hypothesized that CIE procedures induced a change in CB1 receptor function. We applied a CB1 receptor agonist (1 µM WIN 55,212–2) (*Figure 2E,F*), but found a similar decrease in EPSC amplitude between Air (44.72 ± 1.98% of baseline at 30–40 min, n = 6, paired t-test vs. baseline: $t_5$ = 27.96, p<0.0001) and CIE-exposed mice (47.08 ± 9.03% of baseline at 30–40 min, n = 6, paired t-test vs. baseline: $t_5$ = 5.86, p<0.01) (unpaired t-test comparing Air and CIE mice: $t_{10}$ = 0.25, p=0.80).

Another possibility for the observed decrease in OFC transmission may involve a CIE-induced change in the expression of eCB-mediated plasticity (*DePoy et al., 2013*). Given the decreased OFC transmission induced by CIE, we hypothesized that eCB-mediated long-term depression (LTD) would be occluded in CIE exposed mice. eCB-LTD has been extensively studied and described using a high-frequency stimulation (HFS) induction protocol. However, we needed to probe input-specific contributions, and conventional ChR2 shows a strong inactivation and desensitization to HFS (*Lin, 2011*). Thus, we opted to use a group 1 mGluR agonist, DHPG, to induce LTD (*Kreitzer and Malenka, 2007*; *Kreitzer and Malenka, 2005*; *Wu et al., 2015*; *Yin et al., 2006*). A 10 min bath application of DHPG (50 µM) paired with postsynaptic depolarization to 0 mV, resulted in LTD of optically evoked EPSCs (Air: 54.60 ± 8.40% of baseline at 30–40 min, n = 6, paired t-test vs. baseline: $t_5$ = 5.41, p<0.01; CIE: 46.68 ± 4.97% of baseline at 30–40 min, n = 5, paired t-test vs. baseline:

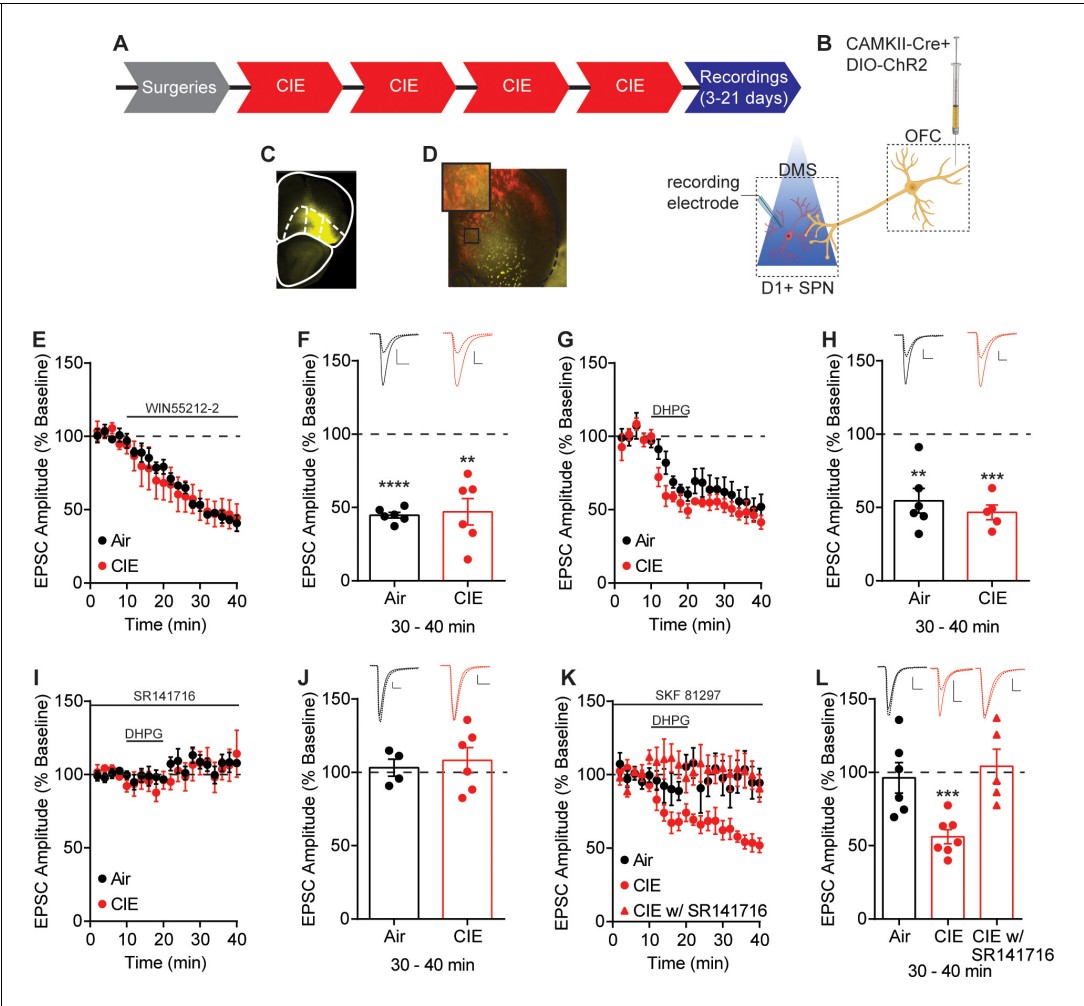

**Figure 2.** Endocannabinoid-mediated plasticity of OFC transmission to D1 SPNs in Air and CIE mice. (**A**) Experimental timeline that includes surgeries followed by four cycles of CIE exposure and whole-cell recordings 3–21 days in withdrawal. (**B**) Schematic of viral injections in OFC and optical stimulation of OFC terminals during whole-cell recordings of D1 SPNs in the DS. (**C**) Representative viral expression of ChR2 in OFC. (**D**) Example ChR2 expression of OFC terminals and expression of tdTomato in D1 SPNs in the DS. (**E**) Bath application of a CB1 receptor agonist, WIN55212-2 (1 µM), during optical stimulation of OFC terminals to D1 SPNs in Air (n = 6 cells, four mice) and CIE (n = 6 cells, three mice). (**F**) Bar graphs representing the percentage change from baseline (min 0–10) after bath application of WIN55212-2 (min 30–40). (**G**) Endocannabinoid mediated long-term depression induced by bath application of a group 1 mGluR agonist, DHPG (50 µM) paired with depolarization to 0 mV in Air (n = 6 cells, three mice) and CIE (n = 5 cells, three mice). (**H**) Bar graphs representing the percentage change from baseline after bath application of DHPG. (**I**) DHPG-LTD of OFC transmission is blocked by a CB1 receptor antagonist, SR141716 (1 µM) in both Air (n = 4 cells, three mice) and CIE (n = 6 cells, three mice). (**J**) Bar graphs representing the percentage change from baseline after bath application of DHPG in the presence of SR141716. (**K**) D1 agonist (3 µM SKF 81297) blocks the expression of DHPG-LTD in Air mice (n = 6 cells, four mice) but has no effect in mice exposed to CIE (n = 7 cells, five mice; w/ SR141716 n = 5 cells, two mice). (**L**) Bar graphs representing the percentage change from baseline after bath application of DHPG in the presence of SKF 81297 or SKF 81297 with SR141716. Scale bars represent 10 ms (horizontal) and 50 pA (vertical). Data points and bar graphs represent mean ± SEM. Bonferroni-corrected or paired (vs. baseline) two-tailed t-test, **p≤0.01, ***p≤0.001, ****p≤0.0001.

The online version of this article includes the following source data and figure supplement(s) for figure 2:

**Source data 1.** Endocannabinoid-mediated plasticity of OFC transmission to D1 SPNs in Air and CIE mice source data.

**Figure supplement 1.** No difference between Air and CIE in the expression of DHPG-LTD of electrically evoked EPSCs.

**Figure supplement 1—source data 1.** No difference betwen Air and CIE in the expression of DHPG-LTD of electrically evoked EPSCs.

$t_4$ = 10.74, p<0.001) (**Figure 2G**). However, this DHPG-induced LTD was of similar magnitude between Air and CIE-exposed mice (unpaired t-test: $t_9$ = 0.77, p=0.46) (**Figure 2H**). Bath application of a CB1 receptor antagonist, SR141716 (1 µM) showed the observed LTD to be CB1 receptor dependent in both groups (Air: 103.2 ± 5.85% of baseline at 30–40 min, n = 4, paired t-test vs.

baseline: $t_3 = 0.54$, p=0.63; CIE: $108.3 \pm 5.85\%$ of baseline at 30–40 min, n = 6 paired t-test vs. baseline: $t_5 = 0.96$, p=0.38) (*Figure 2I,J*). Thus, CB1 receptor-dependent eCB-LTD at OFC-D1 SPNs synapses also remains intact after CIE exposure.

While CB1 receptor function and eCB-mediated plasticity at OFC-D1 SPN synapses was still intact in CIE mice, we wondered whether regulation of eCB signaling may be disrupted following CIE. Prior work in striatum has shown that activation of D1 receptors prevents the expression of eCB-LTD (*Trusel et al., 2015*; *Wu et al., 2015*). Additionally, positive-timing induction protocols in striatum require D1 receptor blockade for the expression of CB1 receptor dependent LTD (*Shen et al., 2008*). Based on these previous findings, we reasoned that D1 receptor activation may negatively regulate eCB-LTD. We found that activation of D1 receptors by bath application of SKF 81297 (3 μM) in Air controls blocked DHPG-LTD ($96.32 \pm 10.41\%$ of baseline at 30–40 min, n = 6, paired t-test vs. baseline: $t_5 = 0.35$, p=0.74) but had no effect in CIE-exposed mice ($56.12 \pm 4.85\%$ of baseline at 30–40 min, n = 7, paired t-test vs. baseline: $t_6 = 9.06$, p<0.001). The LTD that CIE mice continued to express was still CB1 receptor dependent, as bath application of SR141716 (1 μM) blocked the DHPG- LTD ($104.2 \pm 11.55\%$ baseline at 30–40 min, n = 5, paired t-test vs. baseline: $t_4 = 0.37$, p=0.73) (*Figure 2K,L*).

This effect may not be specific to OFC-D1 SPNs and could also be present at any glutamatergic input onto D1 SPNs. However, evoked EPSCs by electrical stimulation in a subset of mice where OFC-DS transmission was optically assessed, showed a similar magnitude of DHPG-LTD in both Air and CIE-exposed mice that was CB1 receptor dependent (*Figure 2—figure supplement 1A–C*). Furthermore, DHPG-LTD of electrically evoked EPSCs was blocked by SKF 81297 in Air mice (*Figure 2—figure supplement 1F,G*). While activation of D1 receptors with bath application of DHPG did result in a short-term depression in CIE, electrically evoked EPSC amplitudes returned to baseline at 30–40 min (*Figure 2—figure supplement 1F,G*) in contrast to the long-lasting effect observed when optically evoking EPSCs (*Figure 2K,L*). Additionally, we found that this short-term depression was sensitive to a CB1 receptor antagonist, as we observed no short- or long-term changes in EPSC amplitude in the presence of SR141716 (1 μM) (*Figure 2—figure supplement 1F,G*). Thus, the disrupted D1 receptor regulation of long-term eCB plasticity appears to be at least somewhat selective to OFC-D1 SPNs, as we do not see similar changes when examining excitatory input generally. These results raise the hypothesis that D1 receptor signaling may be altered in CIE-exposed mice.

## Chronic alcohol exposure disrupts D1 receptor function

D1 receptors may negatively modulate eCB signaling as activation of D1 receptors increases spontaneous presynaptic neurotransmitter release (*André et al., 2010*), decreases eCB content (*Patel et al., 2003*), and prevents the expression of eCB-LTD (*Wu et al., 2015*). To determine whether CIE alters D1 receptor activity, we used both an in vitro and in vivo assay of D1 receptor function. First, D1 receptor activation in vitro has been shown to increase excitability of D1 SPNs (*Hernández-López et al., 1997*; *Planert et al., 2013*; *Surmeier et al., 2007*). Similar to previous findings, we found that washing on D1 agonist, SKF 81297 (10 μM), enhanced D1 SPN action potential firing in response to current injections in the DS of Air controls (repeated-measures ANOVA: interaction (Current × SKF 81297): $F_{(10, 90)}=5.72$, p<0.0001; main effect of Current: $F_{(10, 90)}=22.46$, p<0.0001; main effect of SKF 81297: $F_{(1, 9)}=32.37$, p=0.0003) (*Figure 3A*). In contrast, bath application of SKF 81297 had no effect on D1 SPN excitability in CIE mice (repeated-measures ANOVA: no interaction [Current × SKF 81297]: $F_{(10, 80)}=0.28$, p=0.98; main effect of Current: $F_{(10, 80)}=29.87$, p<0.0001, but no main effect of SKF 81297: $F_{(1, 8)}=0.09$, p=0.77) (*Figure 3B*). This suggests that D1 receptor signaling is disrupted in CIE mice and is in line with our previous data in which a D1 agonist failed to modulate DHPG-LTD in CIE mice (*Figure 2K,L*).

To test whether CIE alters the function of DS D1 receptors in vivo, we relied on previous findings where an imbalance in striatal dopamine induced rotational behavior (*Glick et al., 1976*; *Martín et al., 2008*). We gave Air and CIE mice a unilateral microinjection of the D1 agonist, SKF 81297 (300 nL, 5 μg/μl), into the left hemisphere of the DS (*Figure 3C*, *Figure 3—figure supplement 1*), and we measured the subsequent number of clockwise (CW) and counterclockwise (CCW) rotations. The prediction is that an increase in unilateral dopamine signaling should normally lead to an increase in CW rotational behavior (*Figure 3C*). While Air mice showed the predicted increase in CW rotations, CIE mice showed reduced D1-agonist induced rotational behavior. A one-way ANOVA ($F_{(3, 24)}=9.31$, p<0.001) showed that Air controls had significantly more CW rotations than

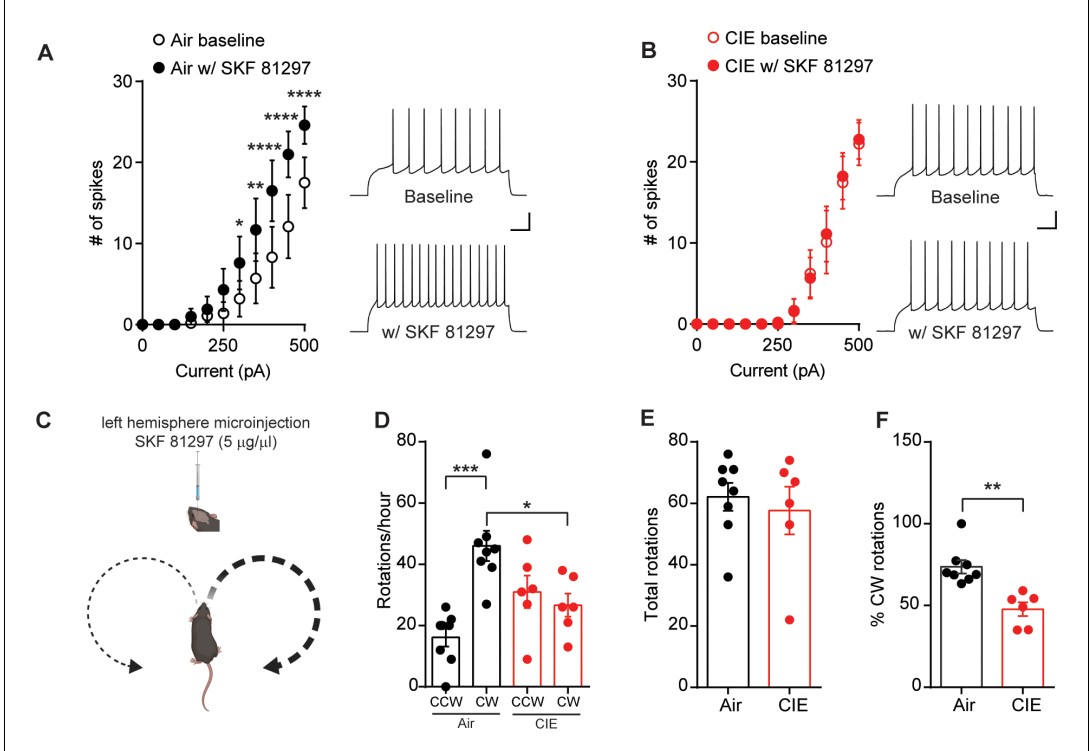

**Figure 3.** D1 receptor function is disrupted in CIE-exposed mice. (**A**) Number of action potentials plotted against current injected under baseline conditions and in the presence of D1 agonist, SKF81297 (10 µM) (left) and sample traces (right) in D1 SPNs of Air (n = 10 cells, three mice) and (**B**) CIE (n = 9 cells, four mice). Scale bars represent 100 ms (horizontal) and 20 mV (vertical). (**C**) Schematic of unilateral microinjections (300 nL) of SKF 81297 (5 µg/µl) in the left hemisphere in the DS and predicted bias toward clockwise (CW) rotations. (**D**) Number of CW rotations and counterclockwise (CCW) rotations in Air (n = 8 mice) and CIE (n = 6 mice) (two vapor cohorts) mice counted over an hour session. (**E**) Total rotations (CW+CCW) in Air and CIE mice. (**F**) Percentage of CW turns in Air and CIE mice. Data points and bar graphs represent mean ± SEM. Bonferroni-corrected post hoc test of repeated-measures ANOVA or unpaired two-tailed t-test, *p≤0.05, **p≤0.01, ***p≤0.001, ****p≤0.0001.

The online version of this article includes the following source data and figure supplement(s) for figure 3:

**Source data 1.** D1 receptor function is disrupted in CIE-exposed mice source data.
**Figure supplement 1.** Unilateral D1 agonist injection and distance traveled.
**Figure supplement 1—source data 1.** Unilateral D1 agonist injection and distance traveled source data.
**Figure supplement 2.** No change in DS D1 or D2 receptor expression in CIE mice compared to Air controls.
**Figure supplement 2—source data 1.** No change in D2 D1 or D2 receptor expression in CIE mice compared to Air controls source data.

CCW rotations (Bonferroni corrected, p<0.001) and had more CW rotations compared to CIE mice (Bonferroni corrected, p<0.05) (*Figure 3D*). Importantly, there was no difference in the total number of rotations between Air and CIE mice (Air: 62.13 ± 4.54, n = 8; CIE: 57.67 ± 7.76, n = 6; unpaired t-test: $t_{12}$ = 0.53, p=0.61) (*Figure 3E*). However, the percentage of CW rotations out of total rotations in CIE mice was significantly lower compared to Air controls (Air: 73.72 ± 4.08, n = 8; CIE: 47.72 ± 4.21, n = 6; unpaired t-test: $t_{12}$ = 4.36, p<0.001) (*Figure 3F*). The blunted D1 receptor agonist response in CIE mice could be due to alterations in D1 receptor expression; however, Western blot analysis showed no change in D1 receptor expression in the DS between CIE and Air mice (*Figure 3—figure supplement 2*). Together, these data provide evidence that CIE results in disruption of D1 receptor function downstream of D1 receptor activation.

## Chronic alcohol exposure enhances depolarization-induced suppression of excitation of OFC transmission to D1 SPNs

Given the chronic alcohol exposure-induced long-lasting disruption of D1 receptor function and the role of D1 receptors in the negative modulation of eCB signaling (*André et al., 2010*; *Patel et al., 2003*), we hypothesized that we may see an increase in eCB signaling in CIE mice. To test for CIE-

induced changes in eCB signaling in OFC-D1 SPN transmission, we used depolarization-induced suppression of excitation (DSE), a short-term form of eCB plasticity (*Diana and Marty, 2004*). To induce DSE of optically evoked EPSCs from OFC terminals (*Figure 4A*), D1 SPNs were depolarized to +50 mV for 10 s which resulted in a short-term decrease of EPSC amplitudes in both Air (DSE Air: $25.78 \pm 3.82$, n = 8, three mice, paired t-test vs. baseline: $t_7 = 6.75$, p<0.001) and CIE mice (DSE CIE: $42.10 \pm 8.25$, n = 9, three mice, paired t-test vs. baseline: $t_8 = 15.31$, p<0.0001) that was CB1 receptor dependent (bath application of SR141716 [1 µM], DSE Air: $2.71 \pm 3.09$, n = 5, paired t-test vs. baseline: $t_4 = 0.88$, p=0.43; DSE CIE: $1.77 \pm 5.20$, n = 6, paired t-test vs. baseline: $t_5 = 0.34$, p=0.75) (*Figure 4B,C*). We found that CIE mice showed a larger magnitude of DSE than did Air controls (unpaired t-test Air vs. CIE: $t_{15} = 3.53$, p<0.01), indicating a CIE-induced increase in eCB signaling in OFC-D1 SPN transmission (*Figure 4B,C*). To again determine whether this effect is selective for OFC-D1 input, we examined DSE in electrically evoked EPSCs (*Figure 4D*) and found DSE in both Air (DSE Air: $32.96 \pm 2.50$, n = 6, three mice, paired t-test vs. baseline: $t_5 = 13.19$, p<0.0001) and CIE mice (DSE CIE: $36.84 \pm 7.00$, n = 6, three mice, paired t-test vs. baseline: $t_5 = 5.27$, p<0.01), that was also sensitive to CB1 receptor blockade (DSE Air: $2.40 \pm 4.88$, n = 6, paired t-test vs.

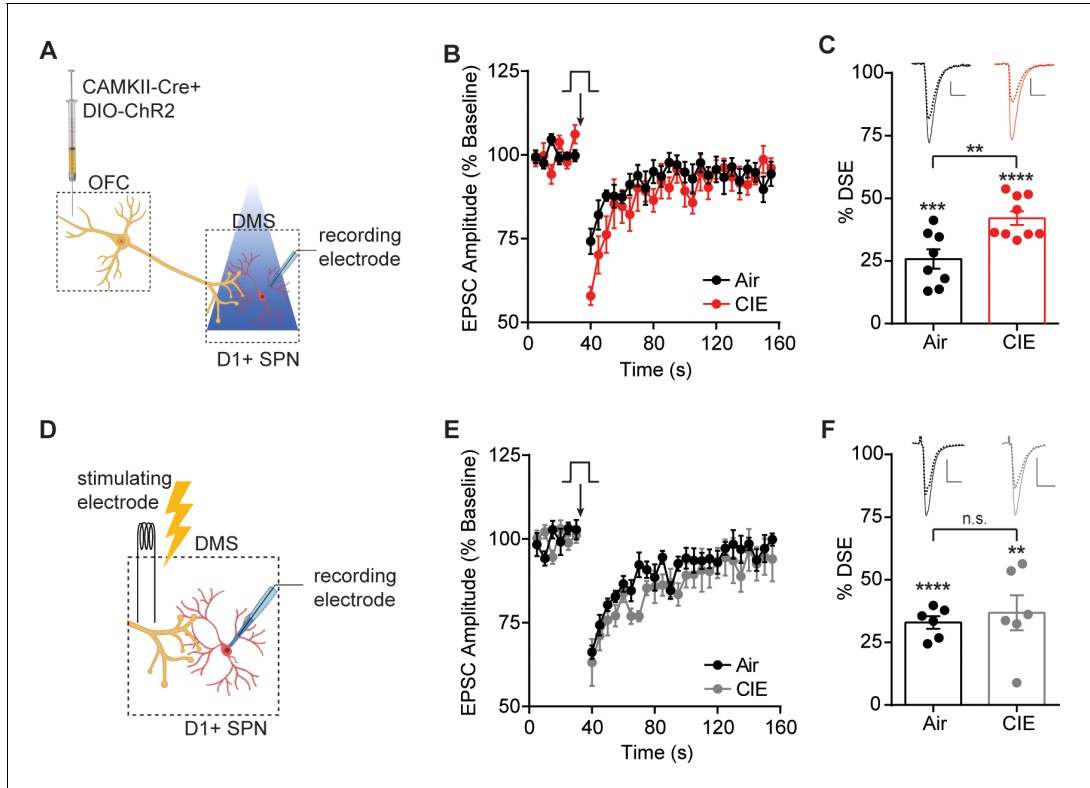

**Figure 4.** CIE-induced enhancement of DSE at OFC terminals to D1 SPNs in the DS. (**A**) Schematic of viral injections in OFC and optical stimulation of OFC terminals during whole-cell recordings of D1 SPNs in the DS. (**B**) Depolarization-induced suppression of excitation (DSE) of OFC inputs to D1 SPNs in Air (n = 8 cells, three mice) and CIE (n = 9 cells, three mice). (**C**) Bar graphs representing the percentage change from baseline immediately after depolarization. (**D**) Schematic of local electrical stimulation during whole-cell recordings of D1 SPNs in the DS. (**E**) DSE of excitatory inputs to D1 SPNs using electrical stimulation in Air (n = 6 cells, two mice) and CIE (n = 6 cells, three mice). (**F**) Bar graphs representing the percentage change from baseline immediately after depolarization. Scale bars represent 10 ms (horizontal) and 50 pA (vertical). Data points and bar graphs represent mean ± SEM. Scale bars represent 10 ms (horizontal) and 50 pA (vertical). Unpaired (Air vs. CIE) or paired (vs. baseline) two-tailed t-test, **p≤0.01, ***p≤0.001, ****p≤0.0001.

The online version of this article includes the following source data and figure supplement(s) for figure 4:

**Source data 1.** CIE-induced enhancement of DSE at OFC terminals to D1 SPNs in the DS source data.

**Figure supplement 1.** DSE in D2 SPNs and effect of SR141716 on DSE in D1 SPNs.

**Figure supplement 1—source data 1.** DSE in D2 SPNs and effect of SR141716 on DSE in D1 SPNs source data.

**Figure supplement 2.** CIE-induced effects on D1 receptor function and DSE across recording days.

**Figure supplement 2—source data 1.** CIE-induced effects on D1 receptor function and DSE across recording days source data.

baseline: $t_5 = 0.49$, p=0.64; DSE CIE: $3.97 \pm 1.94$, n = 5, paired t-test vs. baseline: $t_4 = 2.05$, p=0.11). In contrast to optically evoked EPSCs at OFC-D1 SPNs synapses (*Figure 4A–C*), Air and CIE mice showed a similar magnitude of DSE when EPSCs were electrically evoked (unpaired t-test Air vs. CIE: $t_{10} = 0.52$, p=0.61) (*Figure 4D–F*). Furthermore, the CIE-induced increase in DSE was cell-type selective as we observed no difference in DSE of D2 SPNs of Air and CIE mice with either optical stimulation of OFC terminals or local electrical stimulation (*Figure 4—figure supplement 1*). The CIE-induced increase in DSE was seen throughout the protracted withdrawal period (*Figure 4—figure supplement 2*). These results suggest that the increase in DSE of OFC transmission onto D1 SPNs following chronic alcohol exposure is at least partially selective to the OFC input.

## CB1 receptor antagonist restores OFC transmission and goal-directed control

The above data suggests an enhancement of eCB signaling at OFC-D1 SPN synapses in CIE mice. Given this, we reasoned that CB1 receptor antagonism may alter the aberrant in vivo activity seen in OFC-DS terminals and restore sensitivity to outcome devaluation. However, the prevalence of CB1 receptors in cortico-basal ganglia circuits makes indirect actions on circuit activity also likely, which could obscure or counter effects at OFC-DS terminals. To examine this, we gave systemic injections of Vehicle or CB1 antagonist, SR141716 (3 mg/kg), prior to operant sessions and measured subsequent calcium activity in OFC-DS terminals during behavior (*Figure 5A*). Similar to prior findings that blocking CB1 receptors does not interfere with the ability to lever press (*Hilário et al., 2007*), we also found that systemic administration of SR141716 had no obvious effect on gross lever press-related behavior (*Figure 5—figure supplement 1*).

We first looked at terminal activity during lever press bouts, where we previously saw slightly enhanced activity in CIE mice (*Figure 1N,O*). In Air control mice, systemically blocking CB1 receptors slightly decreased OFC terminal activity during lever press bouts (FDR-corrected two-sided permutation test, all event trials: Vehicle: n = 462; SR141716: n = 359; session average: Vehicle: n = 7; SR141716: n = 7) (*Figure 5B,D*). This was a significant decrease in Net AUC with all event trials (Vehicle: $-0.21 \pm 0.12$, n = 462; SR141716: $-1.42 \pm 0.16$, n = 359; unpaired t-test: $t_{819} = 6.27$, p<0.0001) (*Figure 5C*), but not session average (paired t-test: $t_6 = 1.64$, p=0.15) (*Figure 5E*). Similarly, CIE mice also showed a slight decrease in OFC terminal activity prior to lever press bout onset with CB1 receptor blockade (FDR-corrected two-sided permutation test, all event trials: Vehicle: n = 536; SR141716: n = 385; session average: Vehicle: n = 7; SR141716: n = 7) (*Figure 5F,H*). However, this did not lead to a significant difference in Net AUC analyses with all event trials (Vehicle: $0.04 \pm 0.09$, n = 536; SR141716: $-0.17 \pm 0.14$, n = 385; unpaired t-test: $t_{919} = 1.50$, p=0.14) (*Figure 5G*), although there was a trend when examining session average per mouse (paired t-test: $t_6 = 2.24$, p=0.06) (*Figure 5I*). As blocking CB1 receptors at OFC-DS terminals should prevent any eCB-related decrease in calcium activity (*Kreitzer and Regehr, 2001*), this suggests the slightly reduced lever-press-related OFC-DS terminal activity is due to eCB actions on broader circuit activity.

However, eCBs are thought to be released at glutamatergic synapses in DS in an activity-dependent manner (*Adermark et al., 2009*; *Adermark and Lovinger, 2009*), and we saw larger increases in OFC-DS terminal activity during periods of associated with outcome retrieval. Hence, we also examined effects of blocking CB1 receptors on the reduced outcome retrieval activity observed in Air and CIE mice (*Figure 1P,Q*). Systemic administration of the CB1 antagonist in Air controls resulted in a brief, slight increase OFC terminal activity prior to the first head entry after reinforcer delivery (FDR-corrected two-sided permutation test, all event trials: Vehicle: n = 177; SR141716: n = 162; session average: Vehicle: n = 8; SR141716: n = 8) (*Figure 5J,L*), but this was not sufficient to induce a difference in Net AUC examining all event trials (Vehicle: $1.05 \pm 0.28$, n = 177; SR141716: $1.15 \pm 0.32$, n = 162; unpaired t-test: $t_{337} = 0.24$, p=0.81) (*Figure 5K*) or session average per mouse (paired t-test: $t_7 = 0.64$, p=0.54) (*Figure 5M*). In contrast, systemic administration of the CB1 antagonist in CIE mice produced a significant and sustained increase in OFC terminal activity during this period of outcome retrieval (FDR-corrected two-sided permutation test, all event trials: Vehicle: n = 275; SR141716: n = 219; session average: Vehicle: n = 10; SR141716: n = 10) (*Figure 5N,P*). This was also reflected in the significant increase in net AUC with all event trials (Vehicle: $0.09 \pm 0.18$, n = 275; SR141716: $0.83 \pm 0.16$, n = 219; unpaired t-test: $t_{492} = 3.10$, p<0.01) (*Figure 5O*) and with session average per mouse (paired t-test: $t_9 = 2.22$, p=0.054) (*Figure 5Q*). Thus, it appears that systemic administration of a CB1 antagonist produces different effects on OFC-

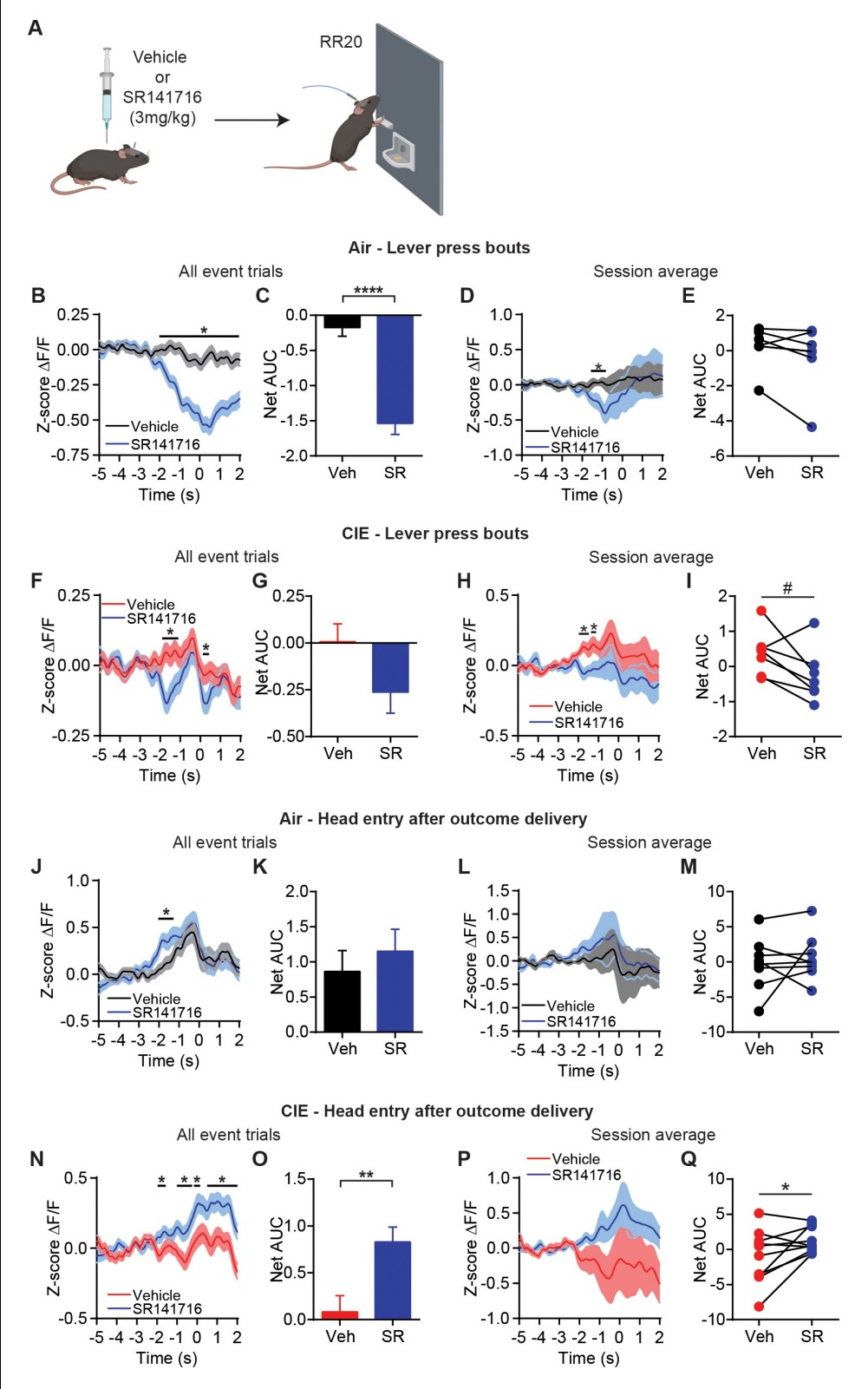

**Figure 5.** CB1 receptor antagonist restores OFC terminal activity for outcome encoding. (**A**) Schematic that depicts injections with vehicle or SR141716 followed by an operant session under a random ratio 20 (RR20) schedule of reinforcement in Air (n = 9 mice, three vapor cohorts) and CIE (n = 11 mice, three vapor cohorts). (**B**) Z-score of ΔF/F GCaMP6s trace recorded in OFC terminals during lever press bouts in mice injected with vehicle

*Figure 5 continued on next page*

*Figure 5 continued*

or SR141716 including all event trials and (C) net area under the curve (AUC) in Air mice. (D) Z-score of ΔF/F
GCaMP6s traces during lever press bouts including session averages and (E) net AUC of Z-score GCaMP6s signal
during lever press bouts in Air mice. (F) Z-score of ΔF/F GCaMP6s trace recorded in OFC terminals during lever
press bouts in mice injected with vehicle or SR141716 including all event trials and (G) net area under the curve
(AUC) in CIE mice. (H) Z-score of ΔF/F GCaMP6s traces during lever press bouts including session averages and (I)
net AUC of Z-score GCaMP6s signal during lever press bouts in CIE mice. (J) Z-score of ΔF/F GCaMP6s trace
recorded in OFC terminals during the first head entry after outcome delivery in mice injected with vehicle or
SR141716 including all event trials and (K) net AUC for each signal in Air mice. (L) Z-score of ΔF/F GCaMP6s trace
recorded in OFC terminals during the first head entry after outcome delivery in mice injected with vehicle or
SR141716 including session averages and (M) net AUC for each signal in Air mice. (N) Z-score of ΔF/F GCaMP6s
trace recorded in OFC terminals during the first head entry after outcome delivery in mice injected with vehicle or
SR141716 including all event trials and (O) net AUC for each signal in CIE mice. (P) Z-score of ΔF/F GCaMP6s trace
recorded in OFC terminals during the first head entry after outcome delivery in mice injected with vehicle or
SR141716 including session averages and (Q) net AUC for each signal in CIE mice. Data points and bar graphs
represent mean ± SEM. Two-sided FDR-corrected permutation test or paired two-tailed t-test, *p≤0.05, **p≤0.01,
****p≤0.0001.

The online version of this article includes the following source data and figure supplement(s) for figure 5:

**Source data 1.** CB1 receptor antagonist restores OFC terminal activity for outcome encoding source data.
**Figure supplement 1.** CB1 antagonist did not alter lever press behavior between training days.
**Figure supplement 1—source data 1.** CB1 antagonist did not alter lever press behavior between training days source data.
**Figure supplement 2.** OFC terminal activity in DS with vehicle or CB1 antagonist.
**Figure supplement 2—source data 1.** OFC terminal activity in DS with vehicle or CB1 antagonist source data.

DS terminal activity dependent on the behavior. Notably, we see a large CB1 receptor antagonist-
induced increase in OFC-DS terminal activity during periods associated with outcome retrieval in CIE
mice, in which OFC-DS terminal is normally blunted and endocannabinoid signaling is enhanced.

Given this, we hypothesized that blocking eCB signaling in DS would restore sensitivity to out-
come devaluation in chronic alcohol-exposed mice. Two separate cohort replicates of mice were
implanted with bilateral cannula targeting DS (*Figure 6—figure supplement 1*) and then exposed
to Air or CIE followed by lever press training for a food outcome (*Figure 6A,B*) as previously
described (*Figure 1*). Air and CIE mice pressed the lever at similar levels (repeated-measures
ANOVA: main effect of Training day: $F_{(8, 240)}=161.5$, p<0.0001; no main effect of CIE exposure or
interaction: Fs < 0.13) (*Figure 6C*) and earned a similar number of outcomes across training
(repeated-measures ANOVA: main effect of Training day: $F_{(8, 240)}=71.45$, p<0.0001; no main effect
of CIE exposure or interaction: Fs < 0.18) (*Figure 6D*). As described above, lever presses were orga-
nized into bouts (*Figure 1*). Similar to the previous cohorts, we found that lever press bouts were dif-
ferentially executed in Air and CIE mice in that CIE mice had slightly less presses per bout, shorter
inter-press intervals, and, consequently, shorter bout durations (*Figure 6—figure supplement 1*).

We then examined the sensitivity of lever pressing to changes in expected outcome value using
outcome devaluation procedures (*Figure 6E*). Just prior to outcome devaluation procedures, mice
were injected with either vehicle or CB1 antagonist, SR141716, directly in the DS. We saw that in Air
mice, intra-DS SR141716 had no effect on outcome devaluation responding. Once again, CIE-
exposed mice treated with vehicle did not show sensitivity to outcome devaluation. However, CIE
mice that were given an injection of SR141716 in the DS significantly decreased lever press respond-
ing on the devalued day relative to the value day (*Figure 6F*). A repeated-measures ANOVA showed
a significant interaction (Training day × Treatment: $F_{(3, 28)}=4.41$, p=0.01) and main effect of Devalua-
tion day ($F_{(1, 28)}=33.28$, p<0.0001) but no main effect of Treatment (F < 0.1). Thus, locally blocking
CB1 receptors in the DS of CIE mice was sufficient to restore value-based decision-making.

## Discussion

Often differing symptoms in psychiatric conditions produce opposing changes in the activity of a
brain area. As there is increasing potential for cortical activity modulation to be used in the treat-
ment of psychiatric disorders, we need a greater understanding of how such cortical modulation will

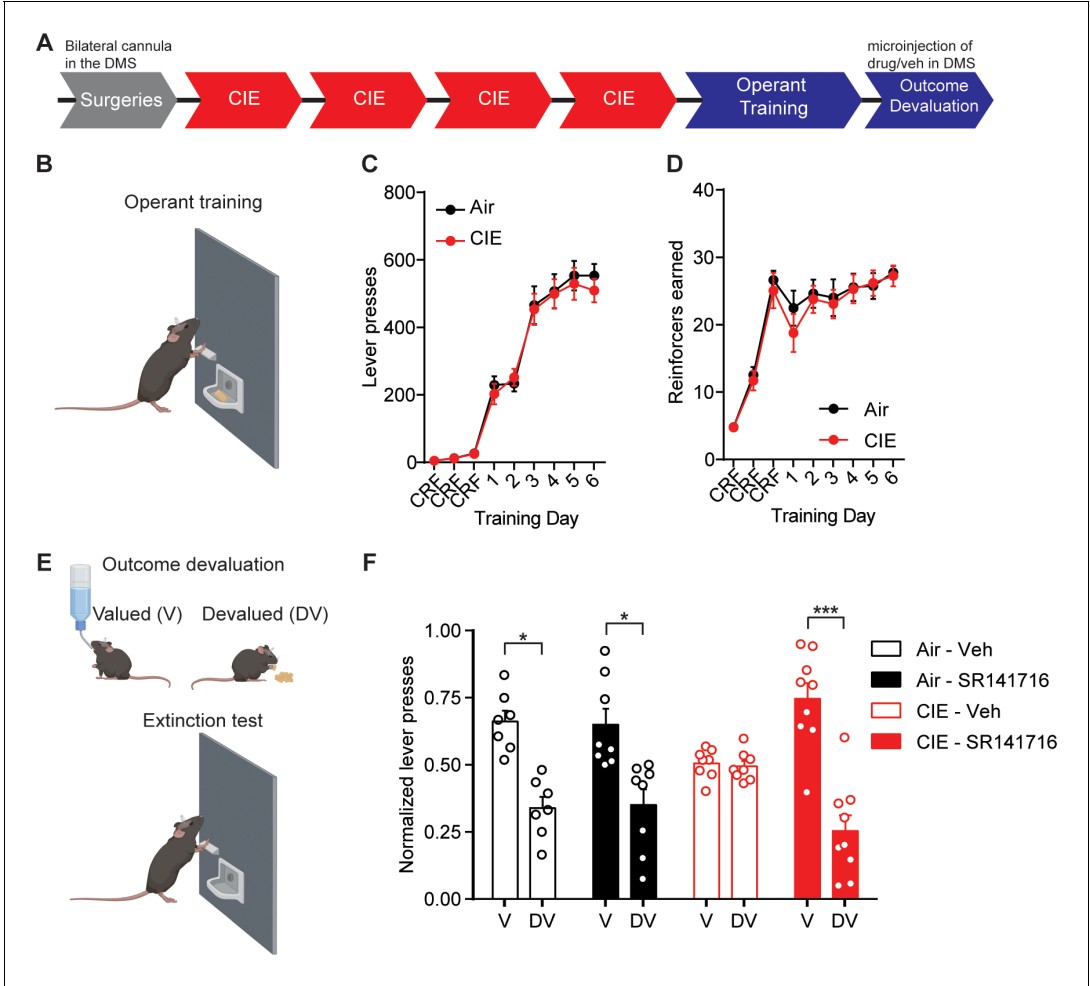

**Figure 6.** Blockade of CB1 receptors in the DS restores value-based decision-making in CIE mice. (**A**) Experimental timeline that includes surgeries followed by four cycles of CIE exposure, operant training and outcome devaluation. (**B**) Schematic of mouse lever press training in which Air (Vehicle: n = 7 mice, SR141716: n = 8 mice, three vapor cohorts) and CIE mice (Vehicle: n = 8 mice, SR141716: n = 9 mice, three vapor cohorts) were trained under a random ratio schedule of reinforcement for a food outcome. (**C**) Lever presses during operant training in Air and CIE mice. (**D**) Outcome earned during operant training in Air and CIE mice. (**E**) Schematic of outcome devaluation procedure that includes 1 hr access to sucrose (valued) or the outcome earned during training (devalued) followed by a 5 min extinction test. (**F**) Normalized lever presses showing the distribution of lever pressing between the valued and devalued day in Air and CIE mice that received microinjections (300 nL) of vehicle or SR141716 (2 μM). Data points and bar graphs represent mean ± SEM. Bonferroni-corrected post hoc test of repeated-measures ANOVA, *p≤0.05, ***p<0.001.

The online version of this article includes the following source data and figure supplement(s) for figure 6:

**Source data 1.** Blockade of CB1 receptors in the DS restores value-based decision-making in CIE mice source data.

**Figure supplement 1.** Behavioral data during lever press training and outcome devaluation in treatment and control groups of Air and CIE mice.

**Figure supplement 1—source data 1.** Behavioral data during lever press training and outcome devaluation in treatment and control groups of Air and CIE mice source data.

---

affect downstream circuit engagement and information representation. Here we find that chronic alcohol exposure enhances activity associated with actions in OFC terminals in striatum, while also reducing activity associated with outcome retrieval. We identify one mechanism potentially responsible for the reduced OFC-DS activity specific to epochs of outcome retrieval: a disruption in D1 receptor function in SPNs and an enhancement of eCB signaling that reduces OFC-D1 SPN transmission in the DS (*Figure 7*). Restoration of OFC-DS transmission through CB1 receptor blockade restored use of reward value to control decision-making. Thus, CIE induces long-lasting, computational-specific changes to cortical transmission in part through cell-type, synapse-specific changes in postsynaptic modulation of this cortical terminal release.

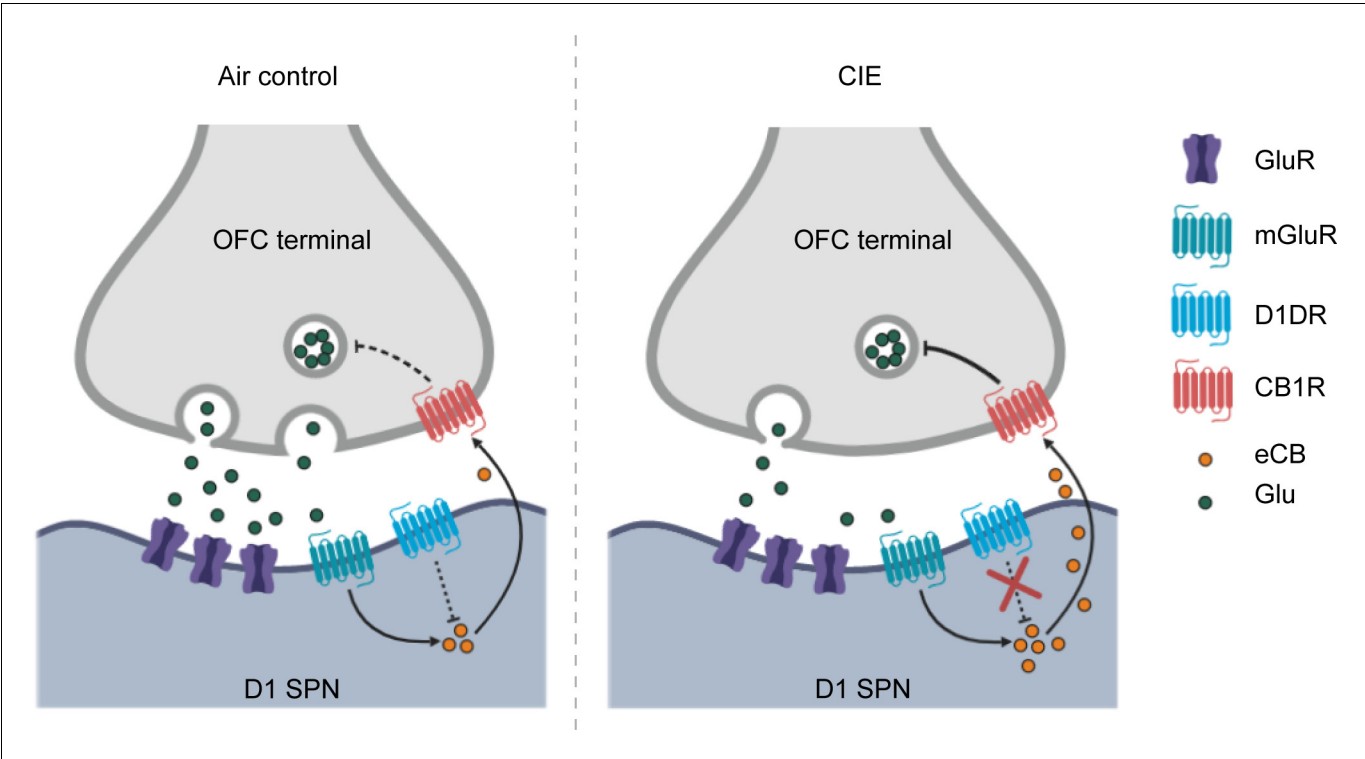

**Figure 7.** Summary of hypothesized alterations in D1 receptor function and endocannabinoid signaling in D1 SPNs of CIE-exposed mice. In Air controls (left), D1 receptor activity regulates endocannabinoid production and/or release. In CIE mice (right), the loss of D1 receptor function leads to unregulated endocannabinoid signaling and the long-term reduction in glutamate release from OFC inputs.

We observed OFC-DS terminal activity profiles reminiscent of common OFC observations in humans: OCD-related hyperactivity (*Milad and Rauch, 2012*; *Pauls et al., 2014*; *Robbins et al., 2019*), the hallmark representation of reward evaluation (*Fellows, 2007*; *Wallis, 2007*), and dependence-induced hypoactivity (*Catafau et al., 1999*; *Lüscher et al., 2020*; *Schoenbaum et al., 2016*; *Schoenbaum et al., 2006*; *Volkow et al., 1997*; *Volkow and Fowler, 1994*). CIE mice showed slightly increased OFC terminal activity during lever press behavior (*Figure 1*), similar to the increased OFC-DS activity reported in compulsive phenotypes (*Pascoli et al., 2018*). While we did not directly measure markers of compulsivity in the present studies, we have previously reported that CIE procedures can increase response rates for both food outcomes (*Renteria et al., 2018*) and alcohol self-administration (*Renteria et al., 2020*). This raises the hypothesis that OFC-DS transmission has a role in alcohol-induced compulsive seeking. However, while this increased OFC terminal activity may drive the increase in response rate in CIE mice, it may not influence reward evaluation processes. We previously observed no correlation between response rate and the sensitivity to outcome devaluation following CIE (*Renteria et al., 2018*). Hence action control and the evaluation of outcome value may be differentially affected by alcohol dependence. That we also found reduced OFC-DS terminal activity in CIE mice during periods associated with outcome retrieval shows how population output activity may be differentially modulated depending on the behavior measured. The observations that chronic alcohol exposure significantly decreased reward-related OFC-DS terminal activity, but slightly increased action-related activity, raise one hypothesis that CIE differentially alters afferent activity into OFC thereby altering the capability of OFC-DS projections to send action and outcome information into basal ganglia circuits. Furthermore, the restoration of OFC terminal activity with a CB1 antagonist during outcome retrieval (*Figure 5*) and rescue of goal-directed control when injected directly in DS (*Figure 6*), suggests there also is additional local synaptic modulation of OFC-DS terminal activity during behavior. Interestingly, we found that OFC-DS terminal activity in Air controls, but not CIE mice, ramps up prior to head entries made after outcome delivery

(*Figure 1P,Q*) and following outcome delivery itself (*Figure 1—figure supplement 1*). It is difficult to disentangle whether this is a predictive response, as mice can readily see into the head entry receptacle within the time frame where this rise occurs. It does suggest future works examining whether OFC-DS activity carries predictive information are warranted.

Our data suggests that CIE induces an increase in eCB signaling at OFC-D1 SPN synapses. While multiple aspects of eCB downstream signaling and plasticity were still intact, including CB1 receptor function and the ability to induce an eCB dependent LTD (*Figure 2*), upstream regulation of eCB signaling by D1 receptors was not. Using ex vivo brain slice preparations from CIE mice, we observed no effect of the D1 agonist SKF 81297 on DHPG-LTD or the excitability of D1 SPNs as previously reported (*Hernández-López et al., 1997*; *Planert et al., 2013*; *Surmeier et al., 2007*) and observed in our controls. In vivo unilateral microinjections of SKF 81297 resulted in rotation behavior that was significantly attenuated in CIE mice and there was no change in D1 receptor expression (*Figure 3—figure supplement 2*). These results suggest that CIE disrupts the signaling cascade downstream of D1 receptor activation. Indeed, acute ethanol has been shown to modulate downstream targets of D1 receptor activation (*Ron and Barak, 2016*; *Ron and Messing, 2013*), including cellular inhibition of specific protein kinase C (PKC) isoforms that constitutively phosphorylate D1 receptors to dampen D1 signaling (*Rex et al., 2008*). Withdrawal was found to decrease PKC isoforms in the BLA (*Christian et al., 2012*), although whether similar mechanisms occur in dorsal striatum is not known. As our measures of D1 receptor function were done 3–21 days in withdrawal from chronic ethanol exposure, this evidence raises one hypothesis that the present disruption in D1 receptor function observed may reflect neuroadaptive compensatory changes in PKC activity. Others have shown, in postmortem brain sections taken from human alcoholics, a downregulation of D1 receptor binding sites in the striatum (*Hirth et al., 2016*). In addition, rats exposed to CIE had an increase of dopamine release in the nucleus accumbens shell, but showed a blunted response to D1 receptor activation 21 days in withdrawal (*Hirth et al., 2016*). While plenty of data suggest that chronic alcohol exposure alters dopamine systems, the changes likely depend on the synapse, cell-type, and circuit.

The loss of D1 receptor function in SPNs of CIE exposed mice may be directly related to the enhancement of eCB signaling (*Figures 4* and *7*). Previous works have shown dopamine and eCB signaling interact in which D1 receptor activation may reduce anandamide (AEA) signaling (*André et al., 2010*; *Patel et al., 2003*) (but see *Giuffrida et al., 1999*), although at what point D1 signaling could interfere with AEA synthesis or release is not clear. However, DSE has widely been shown to be mediated by 2-arachidonoyl glycerol (2-AG), and a CIE-induced increase in 2-AG may contribute to the present effects. Others have shown that habitual alcohol seeking may be mediated by an increase in 2-AG, as an inhibitor, DO34, was found to reduce habitual responding (*Gianessi et al., 2020*). Thus, future work examining the contribution of different eCBs is warranted.

Conversely, the eCB system has been extensively studied in the modulation of dopamine transmission (*Covey et al., 2017*), further underscoring the complex interactions of these signaling systems. For example, as eCBs can reduce dopamine release in nucleus accumbens through reducing cortical drive (*Mateo et al., 2017*), there may also be a CIE-induced decrease in endogenous D1R activation at OFC-D1 SPN synapses. Given the large number of synaptic and molecular targets that chronic ethanol exposure can modulate (*Abrahao et al., 2017*; *Lovinger and Roberto, 2013*; *Roberto and Varodayan, 2017*), the synapse and cell-type-specific alterations in orbitostriatal transmission that we have observed are likely the result of ethanol-induced changes in many neuromodulatory and synaptic signaling systems.

The CIE-induced alterations in eCB-mediated signaling and plasticity presented here were found to be at least, in part, specific to the OFC input. Local electrical stimulation (*Figure 2—figure supplement 1*) likely includes OFC inputs in addition to excitatory inputs from other cortical and limbic regions as well as thalamic nuclei (*Wall et al., 2013*). It is possible that other excitatory inputs to D1 SPNs are differentially modulated by chronic alcohol exposure such that changes at individual inputs may be masked with electrical stimulation. However, enhanced eCB signaling is not expected to apply to thalamostriatal inputs to D1 SPNs, as they lack CB1 receptors at presynaptic terminals (*Wu et al., 2015*). Evidence for a CIE-induced change in other excitatory inputs to D1 SPNs may be supported by the DHPG-induced short-term depression observed with electrical stimulation in the presence of a D1 agonist (*Figure 2—figure supplement 1*). As the observed short-term depression was found to be sensitive to a CB1 receptor antagonist, it likely involves a change in eCB signaling,

although the mechanism may differ from that observed with long-term depression (*Castillo et al., 2012*; *Heifets and Castillo, 2009*; *Kano et al., 2009*; *Lovinger, 2008*).

Interestingly, we found the effects of CB1 antagonist on OFC-DS terminal activity to be dependent on the specific component of decision-making represented. Systemic injection of CB1 receptor antagonist was found to decrease OFC terminal activity in DS during initiation of lever press bouts in both Air and CIE mice (*Figure 5*), suggesting the effect the antagonist exerted was not through activation of CB1 receptors on OFC terminals. One explanation could be a disinhibition of inhibitory signaling onto OFC cell bodies or terminals in DS. CB1 receptors are expressed on GABAergic terminals in cortex which, when blocked with a CB1 antagonist, would allow for inhibition of cortical projection neurons. Additionally, in the striatum, CB1 receptors are expressed not only on cortical terminals, but also on SPNs and GABAergic interneurons (*Hu and Mackie, 2015*), which may provide local regulation of excitatory transmission. In contrast, blocking CB1 receptors had little effect on OFC-DS transmission during outcome retrieval epochs in Air controls but significantly increased activity in CIE mice. These observations suggest that differences in instrumental responding and outcome encoding may result from differentially engaged local regulation of OFC-DS transmission. As prior work showed a dependency of outcome devaluation on the activity of OFC-DS terminals (*Gremel et al., 2016*), it is important to note that our current findings suggest that OFC may pass into striatum information pertaining to more than one type of computation employed during outcome devaluation.

The present mechanistic findings, which are not only synapse, cell-type, and projection specific, but likely also dependent on input engagement or computation performed, demonstrate the complexity of how circuits represent information. A circuit informational approach (*Lovinger and Gremel, 2021Benjamini and Hochberg, 1995*), aimed at identifying the strength and pattern of incoming transmission, recruited plasticity mechanisms, and capabilities of local circuit modulation to affect transmission, increasingly appears necessary in order to understand disease-induced disruptions to behavior. Here we further our understanding of how OFC-DS circuitry regulates decision-making, building off of previous works (*Gremel and Costa, 2013*; *Gremel et al., 2016*) to show this pathway carries both action and reward information. Importantly, we now show that action and reward information can recruit different downstream mechanisms depending on the pattern of activation. However, much work remains to be done to understand the neuroadaptations induced by CIE that result in aberrant decision-making. For example, what does the reduction in transmission from one cortical area mean for SPN output in the DS? Results in the present study focus on the effects of CIE on OFC-D1 SPN transmission in the DS, but the loss of value-based decision-making likely also involves changes to habitual processes for which the neurocircuitry is less understood. Work performed in non-human primates suggests chronic heavy alcohol consumption increases excitatory drive and decreases local inhibition within the putamen or dorsolateral striatum (DLS), an area supporting habitual action control (*Cuzon Carlson et al., 2011*). Thus, it may be that alcohol exposure results in a decreased ability of associative cortical input to control DS output as well as disinhibition of DLS. Furthermore, future work could investigate the time course and potential duration of or change in such effects, as the present study was limited to a 3–21 day window in withdrawal. Elucidating the long-term effects of alcohol dependence on cortico-striatal plasticity will further our understanding of the dependence-induced disruptions to decision-making processes that contribute to continual drug-seeking and taking behaviors.

## Materials and methods

### Mice

B6.Cg-Tg(Drd1a-tdTomato)6Calak/J (*Shuen et al., 2008*) and B6.FVB(Cg)-Tg(Drd1-cre)EY266Gsat/Mmucd mice were used for electrophysiological recordings, and C57BL/6J mice were used for behavioral experiments and fiber photometry. All mouse lines were obtained from Jackson Laboratory, and transgenic mouse lines were bred with C57Bl/6J mice (Jackson Laboratory) for one generation, in-house. Adult (>8 weeks) male and female mice were housed in groups of 2–4, with mouse chow (Purina 5015) and water ad libitum and were kept on a 14 hr light/10 hr dark cycle. Behavioral and physiology experiments were conducted during the light phase of the light cycle. All

experiments were approved by the Institutional Animal Care and Use Committees of the University of California San Diego, and experiments were conducted according to NIH guidelines.

## Viral injections

Mice were anesthetized with isoflurane and were given stereotaxically guided bilateral injections into the OFC (coordinates from Bregma in mm: anterior [A], 2.70; medial [M], ±1.65; ventral [V]: 2.65). D1-tdTomato mice, used for patch clamp recordings, were co-injected with 100 nL AAV5-CamKIIa-GFP-Cre and 100 nL AAV5-Ef1a-DIO-ChR2-eYFP in the OFC. D1-Cre mice received additional bilateral injections of AAV5-Flp-tdTomato in the DS (coordinates from Bregma in mm: anterior [A], 0.5; medial [M],±1.5; ventral [V]: 3.25). The coordinates target the medial DS from the perspective of the dorsal–ventral axis, and more medial DS with respect to the medial versus lateral striatal distinctions. For fiber photometry, C57BL/6J mice received unilateral injections with 500 nL AAV-hSyn-axon-GCaMP6s in the OFC. Viral spread and expression in OFC terminals were assessed by imaging the extent of fluorescence in brain slices (Olympus MVX10).

## Chronic intermittent ethanol exposure and repeated withdrawal

Mice were exposed to four rounds of ethanol vapor or air and repeated withdrawal (*Becker, 1994*; *Becker and Lopez, 2004*; *Griffin et al., 2009*; *Lopez and Becker, 2005*). Each round consisted of 16 hr of vapor exposure followed by an 8 hr withdrawal, repeated for four consecutive days. Ethanol was volatilized by bubbling air through a flask containing 95% ethanol at a rate of 2–3 L/min. The resulting ethanol vapor was combined with a separate air stream to give a total flow rate of approximately 10 L/min, which was delivered to the mice housed in Plexiglas chambers (Plas Labs Inc). Blood ethanol concentrations (BEC) were collected at the end of each round from sentinel mice (mean BEC = 34.7 ± 2.0 mM). No pyrazole or loading ethanol injections were given prior to placement in vapor chambers (*Renteria et al., 2018*).

## Operant behavior

Training was conducted as previously described (*Gremel et al., 2016*; *Gremel and Costa, 2013*; *Renteria et al., 2018*). Two days prior to training, mice were food restricted and maintained at ~ 90% of their baseline body weight throughout training and testing. Mice were placed in sound attenuating operant boxes (Med-Associates) and were trained to press a single lever (left or right) for a food reinforcer (chow pellet). Mice were first trained to retrieve the outcome, in the absence of levers, under a random time schedule (RT60) in which the outcome was delivered on average every 60 s. Mice were then trained on a continuous reinforcement schedule (CRF) in each context in which each lever press produced a single outcome, and the maximum number of reinforcers earned in three daily sessions being 5, 15, and 30, respectively. Following CRF training, mice were trained under a random ratio (RR) schedule of reinforcement. Mice received two days of training in RR10 (on average the 10th lever press produces the outcome), followed by 4 days under RR20. Sessions ended after 30 reinforcers were earned or after 60 min had elapsed. Lever press bouts were analyzed using custom scripts on Matlab (*Renteria, 2021a*, https://github.com/gremellab/Renteria-et-al.-2021-eLife; *Renteria, 2021b*; copy archived at swh:1:rev:1dc7163f31ac46ddfa75-d6a9c9fb8841f25500a2). Briefly, we found the inter-press intervals for all lever presses within a session and calculated the mean. Bouts were separated by inter-press intervals that were greater than the mean inter-press interval. Only bouts of at least three lever presses were included for analysis.

Devaluation testing through sensory-specific satiation was conducted across 2 days: a valued day and a devalued day. Mice were allowed to prefeed ad libitum for 1 hr on the food pellet outcome previously earned by lever pressing (devalued day) or a 20% sucrose solution to control for satiety (valued day). Each day immediately following prefeeding, mice underwent a 5 min extinction test in which the number of lever presses made were recorded, but no outcome was delivered. Lever presses on the valued or devalued day were normalized to total lever presses (valued + devalued) across devaluation testing to account for differences in response rates across individual groups. Mice with optical fiber implants for OFC terminal recordings were retrained after outcome devaluation using RR20 schedule of reinforcement and showed response rates during retraining similar to responding prior to outcome devaluation (*Figure 5—figure supplement 1*).

## Fiber photometry

Mice were injected with AAV-hSynapsin-axon-GCaMP6s in the OFC, and fibers were implanted in the DS (coordinates from Bregma in mm: anterior [A], 0.5; medial [M], −1.5; ventral [V]: 3.25). Fibers were attached for the last 5 days of operant training, beginning on the second day of random ratio training. A blue LED (470 nm, Thorlabs) was used for excitation of OFC terminals in the DS. Fluorescence emissions were collected with a bifurcated fiber (Thorlabs), which allowed for simultaneous recording of two mice. The dual-fiber core was focused through a 10× objective (Olympus) onto a CMOS camera (FLIR Systems). Fluorescence intensity and analog signals for lever press, head entries, and outcome delivery were acquired simultaneously, thresholded, and timestamped for later analyses using Bonsai software (*Lopes et al., 2015*). Photometry and behavioral data were imported to Matlab (Mathworks Inc, Natick, MA) and were analyzed based on prior works (*Markowitz et al., 2018*) using custom scripts (*Renteria, 2021a*, https://github.com/gremellab/Renteria-et-al.-2021-eLife). We did not employ a reference channel in the 405–410 nm wavelength range, as GCaMP6s has an excitation peak within that range. To account for photobleaching and signal decay across a session, we fit a double exponential line on the raw calcium fluorescence signal and normalized the signal to this fit. We then estimated baseline fluorescence (F0), by calculating the running 10th percentile of the fluorescence signal intensity range using a 15 s sliding window for the entire session. ΔF was calculated by subtracting F0 from the fitted signal, and then ΔF was normalized by ΔF/F0. Only sessions in which the 97.5% of ΔF/F0, across the entire session trace (independent of any behavioral response), exceeded a 1% change from baseline fluorescence were included (*Markowitz et al., 2018*). For each session, 'trials' were composed of peri-event signals from −5 to 5 s around event onset. In each 'trial', 50 ms bins were z-scored to the session's mean pre-event baseline (−5 to −2 s). We analyzed this photometry data two ways. First, these z-scored ΔF/F0 traces were combined across all mice within a group. This was done to preserve the variance seen within a subject. Second, we averaged these z-scored ΔF/F0 traces for a given animal session and then averaged these traces across mice within a group. This examines between-mouse variability but does not preserve within-subject variability. Outliers, defined as peak amplitude Z-scores more than three scaled median absolute deviations (above and below) from the median, were removed from group data.

## Brain slice preparation

Mice were at least 12 weeks of age at the time of slice preparation. Coronal slices (250 μm thick) containing the DS were prepared using a Pelco easiSlicer (Ted Pella Inc, Redding, CA). Mice were anesthetized by inhalation of isoflurane and brains were rapidly removed and placed in 4°C oxygenated ACSF containing the following (in mM): 210 sucrose, 26.2 $NaHCO_3$, 1 $NaH_2PO_4$, 2.5 KCl, 11 dextrose, bubbled with 95% $O_2$/5% $CO_2$. Slices were transferred to an ACSF solution for incubation containing the following (in mM): 120 NaCl, 25 $NaHCO_3$, 1.23 $NaH_2PO_4$, 3.3 KCl, 2.4 $MgCl_2$, 1.8 $CaCl_2$, 10 dextrose. Slices were continuously bubbled with 95% $O_2$/5% $CO_2$ at pH 7.4, 32°C and were maintained in this solution for at least 60 min prior to recording.

## Patch clamp electrophysiology

Whole-cell patch clamp recordings were made in identified SPNs in the medial DS. Cells were identified using an Olympus BX51WI microscope mounted on a vibration isolation table. Prior to patching onto a cell, the presence of td-Tomato expression was used to verify cell-type (D1+ or D1−) as well as eYFP expression for terminal expression of ChR2. eYFP expression was never observed in SPN cell bodies. Recordings were made in ACSF containing (in mM): 120 NaCl, 25 $NaHCO_3$, 1.23 $NaH_2PO_4$, 3.3 KCl, 0.9 $MgCl_2$, 2.0 $CaCl_2$, and 10 dextrose, bubbled with 95% $O_2$/5% $CO_2$. ACSF was continuously perfused at a rate of 2.0 ml/min and maintained at a temperature of 32°C. Picrotoxin (50 μM) was included in the recording ACSF to block $GABA_A$ receptor-mediated synaptic currents. Recording electrodes (thin-wall glass, WPI Instruments) were made using a PC-10 puller (Narishige International, Amityville, NY) to yield resistances of 3–6 MΩ. For current-clamp experiments, electrodes were filled with (in mM): 135 $KMeSO_4$, 12 NaCl, 0.5 EGTA, 10 HEPES, 2 Mg-ATP, 0.3 Tris–GTP, 260–270 mOsm (pH 7.3). For voltage clamp experiments, electrodes were filled with (in nM): 120 $CsMeSO_4$, 15 CsCl, 8 NaCl, 10 HEPES, 0.2 EGTA, 10 TEA-Cl, 4 Mg-ATP, 0.3 Na-GTP,

0.1 spermine, and 5 QX-314-Cl. Access resistance was monitored throughout the experiments. Cells in which access resistance varied more than 20% were not included in the analysis.

Glutamatergic afferents were stimulated either electrically or optically. For electrical stimulation, a stainless steel bipolar stimulating electrode (FHC, Inc) was placed dorsal to the recording electrode, about 150–300 μm from the cell body. Optical stimulation was done using 470 nm blue light (4 ms pulse width) delivered via field illumination using a high-power LED (LED4D067, Thor Labs). Light intensity was adjusted to produce optically evoked excitatory postsynaptic currents (EPSCs) with a magnitude of 100–300 pA. Recordings were made using a MultiClamp 700B amplifier (Molecular Devices, Union City, CA), filtered at 2 kHz, digitized at 10 kHz with Instrutech ITC-18 (HEKA Instruments, Bellmore, NY), and displayed and saved using AxographX (Axograph, Sydney, Australia). For long-term depression (LTD) experiments, EPSCs were evoked at 0.05 Hz for a 10 min baseline, followed by a 10 min bath application of (S)−3,5-dihydroxy-phenylglycine (S-DHPG) (50 μM) paired with depolarization to 0 mV. After S-DHPG application, EPSCs were monitored for 20 min at a rate of 0.05 Hz. Data were combined in 2 min bins, and EPSC amplitudes were normalized to the 10 min baseline period. The magnitude of LTD was calculated by averaging normalized EPSC values from 30 to 40 min. For current-clamp recordings, a fixed current was injected for 800 ms in 50 pA steps from −400 pA to +500 pA, and the number of action potentials was counted at each step. Current injections were conducted prior to (baseline) and after a 10–15 min wash on of the D1 receptor agonist SKF 81297 (10 μM). For depolarization-induced suppression of excitation (DSE), EPSCs were evoked at a rate of 0.2 Hz for a 30 s baseline followed by a 10 s depolarization to +50 mV. After depolarization, EPSCs were monitored for 120 s at 0.2 Hz. Data were averaged over three trials for each neuron. EPSC amplitudes were normalized to the average of the baseline period, and the magnitude of DSE was measured as the first point after depolarization. Data from each neuron within a treatment group was combined and presented as mean ± SEM.

## Rotational behavior

To probe for changes in D1 receptor function in vivo, mice were tested over a 3-week period following the conclusion CIE exposure. After cannula-guided injections, mice were placed in a round container, and Bonsai software (*Lopes et al., 2015*) was used for video recording and tracking orientation (radians). The number of rotations was counted using custom code in MATLAB (Mathworks Inc, Natick, MA) in which the difference of orientation at each frame was added until the sum was greater than or equal to 6.28 radians or one full CW rotation. The sum total was reset to 0 with the count of each rotation. CCW turns were counted when the sum was less than or equal to −6.28 radians. Each mouse was tested once for 1 hr and then immediately sacrificed for brain extraction and confirmation of cannula placement.

## Immunoblotting

Tissue punches from the medial DS were prepared using a disposable biopsy puncher (Integra, #3331-A) from fresh-frozen brains of CIE or air-exposed mice and kept at −80°C until further processing. Proteins were extracted in ice-cold lysis buffer containing 150 mM NaCl, 50 mM Tris–HCl, and 1% Triton X-100, pH 7.4 and supplemented with protease inhibitors (Sigma, Complete Mini Protease Inhibitor Cocktail, #11836153001) using a probe tip sonicator (Sonics, Vibra Cell). Homogenates were further incubated on a rotator for 1 hr at 4°C and centrifuged at 12,000 × g for 15 min at 4°C. Supernatant was kept, and protein concentration was measured using the BCA method (Pierce BCA Protein Assay, Thermo Fisher, #23225). Samples were incubated in 2× Laemmli Buffer (120 mM Tris–HCl, pH = 6.8, 20% [w/v] glycerol, 4% sodium dodecyl sulfate, 10% 2-mercaptoethanol, 0.02% bromphenol blue) for 15 min at 50°C, and 5 μg of protein was separated on a 4–20% sodium dodecyl sulfate polyacrylamide gel electrophoresis (Bio-Rad, #4561096) and blotted onto nitrocellulose membranes by wet transfer. Membranes were stained with Ponceau S solution to determine equal protein loading before blocking in 0.1% PBST (phosphate-buffered saline [PBS], 0.1% Tween-20) containing 5% non-fat dry milk for 1 hr at room temperature. Incubation with primary antibodies (guinea pig Anti-D1R 1:1000, Frontier Institute, #2571594; rabbit Anti-D2R 1:1000; Frontier Institute, #2571596; rabbit Anti-cofilin [D3F9] 1:10,000, Cell Signaling Technology, #5175) in blocking buffer was performed overnight at 4°C. Next day, membranes were washed three times in 0.1% PBST and incubated with secondary antibodies (donkey Anti-guinea pig-Alexa-488 and/or donkey Anti-rabbit-

Alexa-647; 1:1000; Jackson Immuno Research) in blocking solution for 2 hr at room temperature before another three 15 min washes in 0.1% PBST. Membranes were briefly rinsed in PBS before imaging using the Bio-Rad ChemiDoc MP imaging system. Band densities (bands at ca. 75–100 kD for D1R or D2R) were quantified using Image J (NIH), normalized to cofilin (loading control; ca. 19 kD), and are expressed as relative intensities. Antibodies for D1 and D2 were used in recent reports and showed decreases in D1R/D2R after CRISPR-mediated knockouts (*Cui et al., 2020*) and showed lack of D1R signal by immunohistochemistry in D1R-KO mice (*Nagatomo et al., 2017*). In addition, the D1R antibody was tested in D1-Cre mice hypomorhpic for D1R and showed considerably less bands at the 70–100 kD range (data not shown).

## Cannula-guided drug injections

Mice were implanted with bilateral cannula (Plastics One) targeting the medial DS ([A], 0.5; [M],±1.5; [V], 3.25) under stereotaxic guidance and were given 1 week of recovery prior to CIE procedures. For rotation behaviors, mice received a unilateral injection (left hemisphere) of 300 nL of SKF81297 (13.49 mM) at a rate of 100 nL/min 20 min prior to testing. For CB1 receptor antagonist effects on devaluation, 300 nL of an 8 µM solution of SR141716A made up of 75% DMSO and 25% saline was injected into each hemisphere 30 min prior to prefeeding for outcome devaluation testing.

## Statistical analysis

Statistical significance was defined as an alpha of $p \leq 0.05$. Statistical analysis was performed using GraphPad Prism 6 (GraphPad Software) and MatLab (Mathworks Inc, Natick, MA). To determine whether there were significant differences in photometry data between groups, permutation testing was performed (*Jean-Richard-Dit-Bressel et al., 2020*; *Pascoli et al., 2018*) in which trials of event-related activity from each treatment group were combined and randomly partitioned 1000 times to compute Z-score means and standard deviations across 50 ms bins. Analysis was limited to $-2$ to 2 s compared to the $-5$ to $-2$ s baseline period. To account for the potential false-discovery rate associated with doing a high number of tests, we applied the Benjamini and Hochberg false discovery rate correction (*Benjamini and Hochberg, 1995*). We also put a significance threshold that required at least four consecutive time points meet the threshold of $p<0.05$. Behavioral and electrophysiology data were analyzed using one-way ANOVA, repeated-measures ANOVA with pre-planned Bonferroni-corrected post hoc analyses or a one- or two-tailed t-test.

## Additional information

### Funding

| Funder | Grant reference number | Author |
|---|---|---|
| National Institutes of Health | AA026077-01A1 | Christina M Gremel |
| National Institutes of Health | F32 AA026776 | Rafael Renteria |
| National Science Foundation | NSF-GRFP DGE-2038238 | Emily T Baltz |
| National Science Foundation | NSF-GRFP DGE-1650112 | Christian Cazares |
| National Institutes of Health | F31 AA027439 | Drew C Schreiner |
| National Institutes of Health | R01DA036612 | Thomas S Hnasko |
| National Institutes of Health | R01NS106822 | Thomas S Hnasko |
| National Institutes of Health | T01BX003759 | Thomas S Hnasko |

The funders had no role in study design, data collection and interpretation, or the decision to submit the work for publication.

### Author contributions

Rafael Renteria, Conceptualization, Formal analysis, Funding acquisition, Investigation, Writing - original draft, Writing - review and editing; Christian Cazares, Ege A Yalcinbas, Formal analysis, Writing - review and editing; Emily T Baltz, Drew C Schreiner, Thomas Steinkellner, Investigation, Writing

- review and editing; Thomas S Hnasko, Resources, Methodology, Writing - review and editing; Christina M Gremel, Conceptualization, Supervision, Funding acquisition, Writing - original draft, Project administration, Writing - review and editing

### Author ORCIDs
Rafael Renteria (iD) http://orcid.org/0000-0002-6199-0293
Christian Cazares (iD) https://orcid.org/0000-0002-8899-2109
Emily T Baltz (iD) https://orcid.org/0000-0001-9770-3666
Ege A Yalcinbas (iD) http://orcid.org/0000-0002-9480-7192
Thomas S Hnasko (iD) http://orcid.org/0000-0001-6176-8513
Christina M Gremel (iD) https://orcid.org/0000-0002-8710-0543

### Ethics
Animal experimentation: This study was performed in strict accordance with the recommendations in the Guide for the Care and Use of Laboratory Animals of the National Institutes of Health. All experiments were approved by the Institutional Animal Care and Use Committees of the University of California San Diego and experiments were conducted according to NIH guidelines.

### Decision letter and Author response
Decision letter https://doi.org/10.7554/eLife.67065.sa1
Author response https://doi.org/10.7554/eLife.67065.sa2

## Additional files

### Supplementary files
• Transparent reporting form

### Data availability
All data generated or analyzed during this study are included in the manuscript and supporting files. Source data for all figures has been provided.

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
