## [Decision Letter]

**Acceptance summary:**

This study examines the mechanisms underlying withdrawal from alcohol dependence and the resulting changes in value-based decision-making, with particular focus on changes to orbitofrontal cortical-dorsal striatal circuitry in the brain. Using fiber photometry, electrophysiology, operant behavior, optogenetics, and pharmacology, the results reveal that these changes arise specifically within spiny projection neurons that express dopamine-type 1 receptors in the striatum in a manner that is mediated by endocannabinoid signaling. Blocking the activity of endocannabinoid receptors was sufficient to rescue value-based decision-making in the alcohol-exposed mice. Overall this work reveals novel information about the mechanisms that lead to cognitive and behavioral dysfunction following withdrawal from alcohol exposure and is likely to be of interest to those interested in alcohol and substance use disorder, reward-seeking behavior, cortico-striatal circuitry, and/or cannabinoid function.

**Decision letter after peer review:**

[Editors’ note: the authors submitted for reconsideration following the decision after peer review. What follows is the decision letter after the first round of review.]

Thank you for submitting your work entitled "Mechanism for differential recruitment of orbitostriatal transmission during outcomes and actions in alcohol dependence" for consideration by *eLife*. Your article has been reviewed by three peer reviewers, including Laura A Bradfield as the Reviewing Editor and Reviewer #1, and the evaluation has been overseen by a Senior Editor.

Our decision has been reached after consultation between the reviewers. Based on these discussions and the individual reviews below, we regret to inform you that your work will currently not be considered for publication in *eLife*. Among other concerns, the reviewers shared considerable concern about the robustness of the photometry data and the long time period over which electrophysiological recordings took place. However, we do feel that there is merit to the manuscript, and would be willing to reconsider a revised version of the manuscript as a new submission if it thoroughly and adequately addresses each reviewer concern as well as the specific points outlined below. Please note that failure to sufficiently address any of the following points upon resubmission will result in rejection without further peer review, so your manuscript may be promptly submitted elsewhere. We certainly understand that you may wish at this point to submit your manuscript elsewhere.

Essential revisions:

1) Please revise language regarding “addiction” and “alcohol dependence” of rats in line with the requests of the reviewers below.

2) The fiber photometry data needs revision and the addition of further analyses. In particular, the issues that must be addressed are:

a) Owing to the concerns described by reviewer 3 and shared by all reviewers, we think it is important to include and discuss the subject averaged data. Please include the subject-averaged data in the main manuscript, along with appropriate quantification, analysis, and Discussion.

b) Please extend the traces to include baseline data for the per trial and per mouse data.

c) Please include a clearer description of how the photometry data were processed in accordance with the requests of reviewer 2.

d) Please include a clearer description of normalisation procedures that were or were not employed.

3) Additional analyses of why fiber photometry activity increased prior to head entry is necessary. Specifically, a comparison of this activity on trials that were unrewarded with trials that were rewarded is necessary to demonstrate that the activity is specific to reward delivery, as claimed by the authors. The authors may wish also to consider whether the sound of the pellet dispenser influenced this signal and/or whether differential behavioral patterns between groups contributed to the discrepant signal.

4) There are still some remaining issues with the length of time over which recordings were taken. To address this, two actions must be taken:

a) In the rebuttal letter, the authors refer to a supplemental figure in their previous publication that demonstrates a lack of evidence for changes over time, please insert a reference to this figure into the manuscript also so that the reader can find it if they wish to do so.

b) Because it is possible that there are differences that occur over time that the current and previous studies may not have detected, please insert a statement into the Discussion noting this possibility, and that future studies may uncover such differences.

5) To ensure transparency and the robustness of the behavioral data, please include the non-normalised (raw) lever press data for the outcome devaluation experiment shown in Figure 1U in the supplement for this figure.*Reviewer #1 (Recommendations for the authors (required)):*

Renteria et al., in this manuscript "mechanism for differential recruitment of orbitostriatal transmission during value and action computations in drug dependence" examine the mechanisms underlying alcohol dependence-induced changes in value-based decision-making with particular focus on orbitofrontal cortical-dorsal striatal circuits. First, they demonstrate that pre-exposing mice to a chronic intermittent ethanol procedure (CIE) leads to differential execution of lever press bouts as well as outcome devaluation performance (which is impaired relative to Air control mice). They then show that CIE induces alterations in OFC terminal activity in the DS, which is in opposite directions during lever press bouts versus during head entries after reinforcer deliveries. They spend the rest of the manuscript unpacking these effects, using both in vivo and in vitro methodologies, demonstrating that: (a) CB1 receptor function and eCB plasticity is not affected by CIE, (b) CIE disrupts D1 receptor function, (c) CIE increases eCB signalling, and this appears to be at least partially selective to the OFC input, and (d) CB1 receptor antagonist restores OFC transmission and outcome devaluation performance, indicating a rescue of goal-directed control.

I have reviewed this manuscript previously, and my reaction to it both then and now is positive. It is a nicely written paper describing an interesting and cohesive body of work. For the most part, my initial concerns have been adequately addressed. I will expand on my remaining concerns below:

1) I suggest that the word “addiction” is replaced with the less stigmatising “substance use disorder”.

2) I still have some concerns that the electrophysiological recordings took place over 3-21 days of withdrawal, which is a relatively long period in which a lot can change. However, I take note of, and appreciate, the author's rebuttal that this is a valid replication of a protracted withdrawal period in adults with alcohol use disorder.

3) I am not sure that all of the other reviewer's concerns have been completely addressed. For example, reviewer #1 previously noted that the increased calcium signal during head entry after reinforcer delivery occur prior to the head entry itself. The authors note that the animal can see into the head entry receptacle and may begin to make its approach prior to the head entry itself being detected, therefore, it is difficult to disentangle whether this is a predictive or reactive response. However, this possibility is not mentioned in the manuscript itself and I think it should be.

With regards to point 2 above I think it would be good to refer to the specific supplementary figures of the authors' prior Nature comms paper in this manuscript as well, however, in order to address the concern that OFC transmission to D1 SPNs may change over this period of time. As multiple reviewers shared this concern, it is probably one that will also occur to readers and should therefore be thoroughly addressed.

I further note that in the rebuttal letter it says that “outcome evaluation” has been changed to “reward evaluation” but this appears to have been changed to “outcome delivery” in the manuscript. This is a change I have previously requested in my review and I fully support, but could be a bit misleading for the other reviewers who were not been involved in the process that led to this change.

Finally, I previously asked the authors to give examples of what neurocognitive disorders could be produced by alcohol use and the authors responded in the rebuttal letter. I should have been clearer that I was asking for this example to be inserted into the text of the manuscript.

Reviewer #2:

The manuscript by Renteria et al., is one I previously reviewed for Biological Psychiatry investigating the role of OFC-dorsal striatal circuits in habitual responding in mice following chronic ethanol inhalation exposure (CIE). The authors have made attempts to address the critiques raised by the previous reviewers and have significantly improved the quality of the manuscript. I do, however, have several additional comments for the authors.

1) Abstract and Introduction should explicit state that experiments were done in alcohol exposed mice rather than the current description of "alcohol dependence" as this suggests the studies were done in humans.

2) The inhalation exposure it thought to model alcohol dependence not to be alcohol dependence (e.g., alcohol dependence in humans does not develop from inhalation exposure). I suggest that the authors revise statements such as "we find that dependence selectively…" to "we find that chronic inhalation exposure to alcohol selectively…". Similar to my first point, just saying alcohol dependence is misleading.

3) The authors argue in their response to reviewers that understanding what the increase in fiber photometry activity prior to head entry reflects is difficult to parse apart in their current paradigm which is not a very satisfying response. Although that may be true, I think the authors need to conduct additional analyses to provide some insight into why the increase is occurring before head entry. For example, if the increase is specific to reward can they show traces for head entries that were not rewarded?

4) I'm still having a difficult time understanding how the fiber photometry data were processed. For example, what does "we then estimated baseline fluorescence by calculating the running 10th percentile of the fluorescence signal using a 15 second sliding window" mean? Why was this used to determine which sessions to include or not? Was the signal normalized to the reference channel? If not, please describe why this was not done as it could be useful for readers and others doing fiber photometry analysis.

Reviewer #3:

The reviewers have responded with additional analyses and set changes to the paper. While the manuscript is improved, there are still some issues that remain unresolved.

1) CIE is not a model of alcohol dependence; mice cannot be diagnosed with mental illness.

2) Photometry data: I am still not convinced the use of each trial in analysis Figure 1 N and P for example are appropriate and truly represent independent samples. Moreover, the per mouse data does not always replicate the trial data, contrary to the authors suggestion in the text. The methods indicate "For each session, "trials" were composed of peri-event signals from -5 to 5 seconds around event onset. In each "trial", 50 ms bins were z-scored to the session's mean pre-event baseline (-5 to -2 seconds)." If this is true, the data for the baseline period should be shown in the figures. This is especially important since the Z score changes are very small in magnitude and examining the noise within the -5 to -2 time frame will allow the readers to more clearly see the changes are event related and not simply noise.

3) The time course was not evaluated as suggested. From the scatter plot shown it clearly looks like DHPG LTD is strongest in late withdrawal days 18-21. While the time window is not an issue per se, the lack of insight into the time dependence of these phenomenon becomes a major limitation with so few data points. More samples at different points would allow for linear regression analysis to show any time-dependence effects.

[Editors’ note: further revisions were suggested prior to acceptance, as described below.]

Thank you for resubmitting your work entitled "Mechanism for differential recruitment of orbitostriatal transmission during actions and outcomes post alcohol exposure" for further consideration by *eLife*. Your revised article has been evaluated by Kate Wassum (Senior Editor) and a Reviewing Editor.

Summary:

This work provides exciting new evidence that chronically exposing mice to alcohol alters their orbitostriatal transmission during the performance of actions and after outcome delivery in decision-making tasks. The authors employ a mixture of techniques including behavioural assessment, fiber photometry, electrophysiological recordings, and pharmacological injections, to comprehensively assess the alterations in activity at orbitostriatal terminals, and to describe the potential mechanism for this disruption as well as how it might be restored. These results provide important insights into the brain mechanisms that underlie disrupted decision-making in alcohol use disorder, a step that is critical for the future development of potential treatments.

The manuscript has been greatly improved and the editors and reviewers appreciate the hard work on the revisions to improve this manuscript. There are a few remaining minor issues that need to be addressed before acceptance, as outlined below:

1) There is some confusion between the "all event trials" and "session average" values (e.g., Figure 1N and 1O). Is the lack of consistency in effect between the "all event trials" and "session average" because a subset of mice had a greater change in dF/F and/or had more bouts than others that when using session averages for each mouse disappear (e.g., all mice were equally represented in the analysis)?

a) To avoid confusion, the following statement in the methods should be moved to or recapitulated in the main text and an explanation of why these measures might differ should be included (e.g., variance within subjects, more events for individual animals, etc) : "We analyzed this photometry data two ways. First, these 675 z-scored ∆F/F0 traces were combined across all mice within a group. This was done to preserve 676 the variance seen within a subject. Second, we averaged these z-scored ∆F/F0 traces for a given 677 animal session, and then averaged these traces across mice within a group. This examines 678 between-mouse variability but does not preserve within-subject variability."

2) In the Results – change alcohol dependence to following CIE or following alcohol inhalation exposure.

3) The antibodies for western blot analysis of D1 and D2 receptors are known to be very problematic (and likely inaccurate measures of protein density). Without confirming the validity of these specific antibodies for assessing D1 and D2 receptor density in mice, including this data in the paper and drawing any conclusions from could be misleading. Since it is not critical to the conclusions, we recommend removing it from the paper.

---

## [Author Response]

[Editors’ note: the authors resubmitted a revised version of the paper for consideration. What follows is the authors’ response to the first round of review.]

Essential revisions:Reviewer #1 (Recommendations for the authors (required)):Renteria et al., in this manuscript "mechanism for differential recruitment of orbitostriatal transmission during value and action computations in drug dependence" examine the mechanisms underlying alcohol dependence-induced changes in value-based decision-making with particular focus on orbitofrontal cortical-dorsal striatal circuits. First, they demonstrate that pre-exposing mice to a chronic intermittent ethanol procedure (CIE) leads to differential execution of lever press bouts as well as outcome devaluation performance (which is impaired relative to Air control mice). They then show that CIE induces alterations in OFC terminal activity in the DS, which is in opposite directions during lever press bouts versus during head entries after reinforcer deliveries. They spend the rest of the manuscript unpacking these effects, using both in vivo and in vitro methodologies, demonstrating that: (a) CB1 receptor function and eCB plasticity is not affected by CIE, (b) CIE disrupts D1 receptor function, (c) CIE increases eCB signalling, and this appears to be at least partially selective to the OFC input, and (d) CB1 receptor antagonist restores OFC transmission and outcome devaluation performance, indicating a rescue of goal-directed control.I have reviewed this manuscript previously, and my reaction to it both then and now is positive. It is a nicely written paper describing an interesting and cohesive body of work. For the most part, my initial concerns have been adequately addressed. I will expand on my remaining concerns below:1) I suggest that the word “addiction” is replaced with the less stigmatising “substance use disorder”.

We have changed “Addiction” to “substance use disorder”.

2) I still have some concerns that the electrophysiological recordings took place over 3-21 days of withdrawal, which is a relatively long period in which a lot can change. However, I take note of, and appreciate, the author's rebuttal that this is a valid replication of a protracted withdrawal period in adults with alcohol use disorder.

We thank the reviewer for that recognition. In regards to including reference to past works and limitations of the current time frame we have added the following to the Results:

“We have previously shown that the decrease in OFC transmission to D1 SPNs persists for up to 21 days in withdrawal (see Supplementary Figure 2 in Renteria et al., 2018).”

The following has been added to the Discussion: “Further, future work could investigate the time course of such effects, as the present study was limited to a 3-21 day window in withdrawal.”

3) I am not sure that all of the other reviewer's concerns have been completely addressed. For example, reviewer #1 previously noted that the increased calcium signal during head entry after reinforcer delivery occur prior to the head entry itself. The authors note that the animal can see into the head entry receptacle and may begin to make its approach prior to the head entry itself being detected, therefore, it is difficult to disentangle whether this is a predictive or reactive response. However, this possibility is not mentioned in the manuscript itself and I think it should be.

We have added the following to the Results presentation as well as to the Discussion:

In the Results, we have added the following: “Since the sound associated with pellet delivery could potentially contribute to activity modulation, we examined OFC-DS terminal activity in relation to outcome delivery (independent of any checking behavior). Again, we found greater modulation within 1 sec of outcome delivery in Air than CIE mice (Figure 1—figure supplement 1D, H). Further, the increase in OFC-DS activity in Air, and to a lesser extent CIE mice, was selective to head entries following outcome delivery when compared to non-rewarded head entries (Figure 1—figure supplement 1E, F). Together, these findings suggest that CIE exposure may lead to slightly enhanced OFC-DS terminal activity associated with actions, but substantially less OFCDS terminal activity during the period associated with outcome perception and/or retrieval. “

In the Discussion, we have added the following: “Interestingly, we found that OFC-DS terminal activity in Air controls, but not CIE mice, ramps up prior to head entries made after outcome delivery (Figure 1P, Q) and following outcome delivery itself (Figure 1—figure supplement 1). It is difficult to disentangle whether this is a predictive response, as mice can readily see into the head entry receptacle within the time frame where this rise occurs. It does suggest future works examining whether OFC-DS activity carries predictive information are warranted.”

With regards to point 2 above I think it would be good to refer to the specific supplementary figures of the authors' prior Nature comms paper in this manuscript as well, however, in order to address the concern that OFC transmission to D1 SPNs may change over this period of time. As multiple reviewers shared this concern, it is probably one that will also occur to readers and should therefore be thoroughly addressed.

The reference has been added, with reference to the specific supplementary figure.

I further note that in the rebuttal letter it says that “outcome evaluation” has been changed to “reward evaluation” but this appears to have been changed to “outcome delivery” in the manuscript. This is a change I have previously requested in my review and I fully support, but could be a bit misleading for the other reviewers who were not been involved in the process that led to this change.

To confirm, we have changed it to outcome delivery.

Finally, I previously asked the authors to give examples of what neurocognitive disorders could be produced by alcohol use and the authors responded in the rebuttal letter. I should have been clearer that I was asking for this example to be inserted into the text of the manuscript.

Our apologies, the text has been changed to: “The DSM-5 categorizes alcohol as one of the few drugs whose dependence can produce neurocognitive disorders such as alcohol-related dementia”.

Reviewer #2:The manuscript by Renteria et al., is one I previously reviewed for Biological Psychiatry investigating the role of OFC-dorsal striatal circuits in habitual responding in mice following chronic ethanol inhalation exposure (CIE). The authors have made attempts to address the critiques raised by the previous reviewers and have significantly improved the quality of the manuscript. I do, however, have several additional comments for the authors.1) Abstract and Introduction should explicit state that experiments were done in alcohol exposed mice rather than the current description of "alcohol dependence" as this suggests the studies were done in humans.

We have changed the text to describe mice that have undergone CIE procedures as “chronic alcohol exposed mice” or “CIE mice” In addition, we have changed the title to “Mechanism for differential recruitment of orbitostriatal transmission during actions and outcomes following chronic alcohol exposure.”

2) The inhalation exposure it thought to model alcohol dependence not to be alcohol dependence (e.g., alcohol dependence in humans does not develop from inhalation exposure). I suggest that the authors revise statements such as "we find that dependence selectively…" to "we find that chronic inhalation exposure to alcohol selectively…". Similar to my first point, just saying alcohol dependence is misleading.

We have changed the text throughout to describe mice that have undergone CIE procedures as “chronic alcohol exposed mice” or “CIE mice”. When we introduce the CIE procedure, we describe it as a model of alcohol dependence.

3) The authors argue in their response to reviewers that understanding what the increase in fiber photometry activity prior to head entry reflects is difficult to parse apart in their current paradigm which is not a very satisfying response. Although that may be true, I think the authors need to conduct additional analyses to provide some insight into why the increase is occurring before head entry. For example, if the increase is specific to reward can they show traces for head entries that were not rewarded?

The photometry graphs labeled “All head entries” included trials of both the first head entry after outcome delivery and all other head entries that were not rewarded. To show that the increase in OFC terminal activity in Air controls is specific to the first head entry after outcome delivery we have conducted the suggested analyses and removed those first head entry after outcome delivery from “All head entries” (now labeled “Non-rewarded head entries”) and find that the increase in terminal activity is absent in both Air and CIE mice (Figure 11—figure supplement E, H).

In addition, we have additional text in Results and Discussion relating to this point.

In the Results, we have added the following: “Even when we examined OFC-DS terminal activity in relation to outcome delivery (independent of any checking behavior), since the sound associated with pellet delivery could contribute to activity modulation, we saw greater modulation within 1 sec of delivery in Air than CIE mice (Figure 1—figure supplement 1D and G). Further, the pattern of OFC-DS activity in Air and CIE mice was absent when we examined head entries excluding ones following outcome delivery (Figure 1—figure supplement 1E and H).”

In the Discussion, we have added the following: “Interestingly, we found that OFC-DS terminal activity in Air controls, but not CIE mice, ramps up prior to head entries made after outcome delivery (Figure 1P, Q) and following outcome delivery itself (Figure 1—figure supplement 1). It is difficult to disentangle whether it is a predictive response, as mice can readily see into the head entry receptacle within the time frame where this rise occurs. It does suggest future works examining whether OFC-DS activity carries predictive information are warranted.”

4) I'm still having a difficult time understanding how the fiber photometry data were processed. For example, what does "we then estimated baseline fluorescence by calculating the running 10th percentile of the fluorescence signal using a 15 second sliding window" mean? Why was this used to determine which sessions to include or not? Was the signal normalized to the reference channel? If not, please describe why this was not done as it could be useful for readers and others doing fiber photometry analysis.

Our apologies. Hopefully we have clarified the text surrounding analyses in both the methods and Results section of our manuscript.

We do not use the commonly used reference channel of 405 nm or 410nm. This is largely because GCaMP6s has an excitation peak within that wavelength range (see example Cheong et al., 2018 and figure from that paper) and does not serve as a true isobestic control. In addition, as lower wavelength light is damaging to neural tissue, we wanted to reduce light exposure over the longer time course of our sessions (1 hour).

Instead, we used the same procedure used by the Sabatini and Datta labs (e.g., Markowitz et al., 2018). To account for photobleaching and signal decay across a session, we first fit a double exponential line on the raw calcium fluorescence signal and normalized the signal to this fit. Then the Baseline fluorescence (F0) was determined from this signal by calculating the running 10^th^ percentile of the fluorescence signal intensity range using a 15 second sliding estimate. This means that the baseline F0 signal is determined by taking a running average (the 15 sec) of the 10% intensity level of the signal (this includes the entire intensity range of trace including peaks, so it is quite a conservative measure). This F0 trace was then subtracted from the fitted signal to get the ∆F trace across the session, and then normalized by ∆F/F0. This is then z-scored. In addition to clarification in the methods, we have added text to the results and figure axis clarifying that the signal is z-scored ∆F/F0.

We also use code from Markowitz et al., in which only sessions where the 97.5% point of the ∆F/F0 exceeded a 1% change from baseline fluorescence (independent of any behavioral response, just across the entire trace for that session) were included. The use of a 97.5% of ∆F/F0 exceeding a 1% change from baseline fluorescence acts as a signal detection mechanism used to determine whether a calcium signal exists anywhere in the entire session trace. In other words, at a minimum, the top 97.5 percentile of your range across your entire session trace has to be at least 1% different than your baseline trace.

Reviewer #3:The reviewers have responded with additional analyses and set changes to the paper. While the manuscript is improved, there are still some issues that remain unresolved.1) CIE is not a model of alcohol dependence; mice cannot be diagnosed with mental illness.

While of course CIE does not capture all aspects of human alcohol use disorder, CIE is a well-established and long-standing rodent model of alcohol dependence in rodents that results in the development of tolerance, withdrawal, and robust increases in voluntary ethanol consumption. Further, CIE was chosen as a model to induce alcohol dependence in rodents by the NIAAA/NIH led Integrative Neuroscience Initiative on Alcoholism (INIA). This NIH organized consortium includes multiple investigators and funded with the goal of understanding brain mechanisms associated with alcohol use disorder RFA-AA-16-005. We have changed our description of mice that have undergone CIE procedures to “chronic alcohol exposed mice” or “CIE mice” so as not to suggest that mice can be diagnosed with mental illness.

2) Photometry data: I am still not convinced the use of each trial in analysis Figure 1 N and P for example are appropriate and truly represent independent samples. Moreover, the per mouse data does not always replicate the trial data, contrary to the authors suggestion in the text. The methods indicate "For each session, "trials" were composed of peri-event signals from -5 to 5 seconds around event onset. In each "trial", 50 ms bins were z-scored to the session's mean pre-event baseline (-5 to -2 seconds)." If this is true, the data for the baseline period should be shown in the figures. This is especially important since the Z score changes are very small in magnitude and examining the noise within the -5 to -2 time frame will allow the readers to more clearly see the changes are event related and not simply noise.

The baseline period of -5 to -2 has been added to the photometry graphs. We also present all events and session average per mouse data side by side in Figure 1 and Figure 5, and associated stats side by side in main text. In Figure 1 we report a slight increase calcium modulation in CIE mice during lever pressing (significant in all events, not in session average), and a much larger calcium modulation associated with head entry after outcome delivery in Air than CIE mice (present in both all events and session average per mouse).

For Figure 5, where we are examining what happens to OFC-DS terminal activity when you give a systemic CB1 antagonist, we find antagonist administration results in a consistent increase in OFC-DS terminal activity during head entry following outcome delivery in CIE but not Air mice.

3) The time course was not evaluated as suggested. From the scatter plot shown it clearly looks like DHPG LTD is strongest in late withdrawal days 18-21. While the time window is not an issue per se, the lack of insight into the time dependence of these phenomenon becomes a major limitation with so few data points. More samples at different points would allow for linear regression analysis to show any time-dependence effects.

Sampling across multiple points in withdrawal to perform linear regression analysis is beyond the scope of this paper. Our goal is not to investigate the effects and time course of protracted withdrawal on OFC-DS transmission but instead to understand how this circuit is altered after the induction of stable and long-lasting alterations in CIE exposed mice. Further, the strong persistence of DHPG-LTD in the presence of a D1 agonist at 21 days, as noted above, suggests that the observed changes are likely stable beyond 21 days. If the observed changes were time dependent and fade across withdrawal, we would expect D1 activation to attenuate DHPG-LTD during late withdrawal, as observed in Air controls. We do think it would be of future interest to map out the emergence of such effect during both exposure and withdrawal timelines and have added discussion to this point.

[Editors’ note: what follows is the authors’ response to the second round of review.]

1) There is some confusion between the "all event trials" and "session average" values (e.g., Figure 1N and 1O). Is the lack of consistency in effect between the "all event trials" and "session average" because a subset of mice had a greater change in dF/F and/or had more bouts than others that when using session averages for each mouse disappear (e.g., all mice were equally represented in the analysis)?

For event types such as lever press bouts, each mouse can have hundreds of bouts per session. For session averages, the variability among those hundreds of bouts is lost. Although some session averages are not significantly different between Air and CIE due to loss of statistical power, such as Figure 1N and 1O, the pattern of the response and net AUC is generally consistent with all event trial data. Differences are not due to different numbers of lever presses between animals, as we see similar distributions of lever press levels/animal within groups.

a) To avoid confusion, the following statement in the methods should be moved to or recapitulated in the main text and an explanation of why these measures might differ should be included (e.g., variance within subjects, more events for individual animals, etc) : "We analyzed this photometry data two ways. First, these 675 z-scored ∆F/F0 traces were combined across all mice within a group. This was done to preserve 676 the variance seen within a subject. Second, we averaged these z-scored ∆F/F0 traces for a given 677 animal session, and then averaged these traces across mice within a group. This examines 678 between-mouse variability but does not preserve within-subject variability."

We have added the following to the main text “Photometry data was analyzed two ways. First, z-scored ΔF/F0 traces were combined across all mice within a group (all event trials). This was done to preserve the variance seen within a subject. Second, we averaged these z-scored ΔF/F0 traces for a given animal session, and then averaged these traces across mice within a group (session average). This examines between-mouse variability but does not preserve within-subject variability.”

2) In the Results – change alcohol dependence to following CIE or following alcohol inhalation exposure.

Alcohol dependence has been changed to “following CIE” and “CIE” respectively.

3) The antibodies for western blot analysis of D1 and D2 receptors are known to be very problematic (and likely inaccurate measures of protein density). Without confirming the validity of these specific antibodies for assessing D1 and D2 receptor density in mice, including this data in the paper and drawing any conclusions from could be misleading. Since it is not critical to the conclusions, we recommend removing it from the paper.

We have chosen to keep the western blot analyses based on the following reasons. The antibodies used for western blot analysis have been knock-out verified by the supplier, Frontiers. Recent papers also used both antibodies for D1 and D2 and showed decreases in D1R/D2R after CRISPR-mediated knockouts (Cui et al., 2020) and show lack of D1R signal by IHC in D1R-KO mice (Nagatomo et al., 2017). In addition, we tested the D1R antibody in D1-Cre mice hypomorhpic for D1R and they showed considerably less bands at the 70-100kD range (data not shown). If the editors disagree after our reasoning above, please let us know and we can then remove it. As stated in communication to the editors, we are hesitant to remove it based on non-specific concerns.